# Enhanced cell deconvolution of peripheral blood using DNA methylation for high-resolution immune profiling

Lucas A. Salas [1], Ze Zhang [1], Devin C. Koestler[2], Rondi A. Butler [3], Helen M. Hansen [4], Annette M. Molinaro[4], John K. Wiencke[4,5], Karl T. Kelsey [3]✉ & Brock C. Christensen [1,6,7]✉

DNA methylation microarrays can be employed to interrogate cell-type composition in complex tissues. Here, we expand reference-based deconvolution of blood DNA methylation to include 12 leukocyte subtypes (neutrophils, eosinophils, basophils, monocytes, naïve and memory B cells, naïve and memory CD4 + and CD8 + T cells, natural killer, and T regulatory cells). Including derived variables, our method provides 56 immune profile variables. The IDOL (IDentifying Optimal Libraries) algorithm was used to identify libraries for deconvolution of DNA methylation data for current and previous platforms. The accuracy of deconvolution estimates obtained using our enhanced libraries was validated using artificial mixtures and whole-blood DNA methylation with known cellular composition from flow cytometry. We applied our libraries to deconvolve cancer, aging, and autoimmune disease datasets. In conclusion, these libraries enable a detailed representation of immune-cell profiles in blood using only DNA and facilitate a standardized, thorough investigation of immune profiles in human health and disease.

[1] Department of Epidemiology, Geisel School of Medicine, Dartmouth College, Lebanon, NH, USA. [2] Department of Biostatistics & Data Science, University of Kansas Medical Center, Kansas City, KS, USA. [3] Departments of Epidemiology and Pathology and Laboratory Medicine, Brown University, Providence, RI, USA. [4] Department of Neurological Surgery, University of California San Francisco, San Francisco, CA, USA. [5] Institute for Human Genetics, University of California San Francisco, San Francisco, CA, USA. [6] Department of Molecular and Systems Biology, Geisel School of Medicine, Dartmouth College, Lebanon, NH, USA. [7] Department of Community and Family Medicine, Geisel School of Medicine, Dartmouth College, Lebanon, NH, USA. ✉email: Karl_Kelsey@brown.edu; Brock.C.Christensen@dartmouth.edu

Advances in DNA methylation microarrays have allowed a greater understanding of how DNA methylation is affected by environmental exposures and altered in chronic diseases[1,2]. Peripheral blood is one of the most common biological matrices for those investigations. Blood DNA methylation profiles include information from multiple cell lineages and, in some cases, cell states. Every cell lineage has unique DNA methylation patterning regulating cell-specific gene expression, and some methods leverage DNA methylation to understand underlying cell heterogeneity[1–7]. The reference-based approach assumes that the principal source of signal variability in a heterogeneous sample (such as blood) reflects the signals' proportions in the different cell components[8]. Constrained projection/quadratic programming (CP/QP) employs purified cell types as reference samples to generate a "reference library," a matrix of differentially methylated sites among cell types, and yields highly accurate estimates of the underlying cell composition in mixed cell populations (e.g., peripheral blood)[9]. Previously established statistical deconvolution frameworks such as CP/QP, support vector regression (CIBERSORT), and robust partial regression (EpiDISH) have similar accuracy and precision in deconvolution estimates[10]. Marker selection methods for library creation use automatic procedures to discern library markers[11] or iterative approaches for selecting sets of markers (IDOL—IDentifying Optimal Libraries) that maximize the accuracy of deconvolution estimates[12]. To enhance the utility of cell-type deconvolution, reference library improvements and expansions of existing libraries to include additional cell types are needed to broaden the scope of DNA methylation-based immune phenotyping[13–16].

Continued methodological advancements are highly dependent on the quality and genome coverage of a reference library. In the original description of CP/QP for methylation-based deconvolution, Houseman et al. developed a library based on an early microarray platform, the Illumina HumanMethylation27k microarray[9]. When the Illumina-HumanMethylation450k technology was released, Jaffe et al. applied the Houseman method with the reference data developed by Reinius et al.[11,17]. They accurately discriminated CD8 and CD4T cells, but NK and granulocytes (neutrophils and eosinophils) discrimination performance showed room for improvement[11,17]. Potentially limiting generalizability, the reference cell populations were purified solely from males of Northern European (Swedish) origin[17,18]. We recently developed a deconvolution library to discriminate better six major cell types (CD4(+) T cells, CD8(+) T cells, NK, B cells, monocytes, and neutrophils) using the Illumina HumanMethylationEPIC technology, hereafter named EPIC IDOL-6[19]. A distinct advantage of this library is the inclusion of more ethnically diverse male and female subjects.

Beyond the six major leukocyte cell types in peripheral blood, there have been further attempts to deconvolve memory and naïve cells and other granulocytes[13,14]. However, they have not been widely adopted or tested as some of the references are not publicly available or involved a combination of different technologies. Some algorithms include other rarer cell subpopulations [e.g., plasmablasts, exhausted CD8(+) T cells] as linearly related scores, but they do not represent the sample's cell-type proportions[15]. Several newborn umbilical cord blood-specific libraries also have been developed[20–24].

Here, we augment reference-based deconvolution of adult peripheral blood DNA methylation data to include memory and naïve cells from cytotoxic and helper T cells and B cells and parse the granulocyte subtypes into neutrophils, eosinophils, and basophils (Fig. 1). Our comprehensive library provides information across 12 different cell subtypes; depending on the hypothesis, categories could be collapsed at in seven additional higher branches (T cells, B cells, CD8T, CD4T, granulocytes, lymphoid, and myeloid), Fig. 1 panel c, resulting in 19 relative cell-type proportions and 19 cell counts (derived data using the complete cell blood counts or flow cytometry). In addition, the library includes multiple cell-derived ratios and proportions, such as the neutrophil to lymphocyte ratio or naïve to memory ratios (see Fig. 1 for 18 known examples), totaling more than 56 total immune profile variables. This library, hereafter named EPIC IDOL-Ext, will find wide application to the study of immune profiles in health and disease. Additionally, a second extended library using probes present on the legacy DNA HumanMethylation450k array, henceforth named 450k IDOL-Ext, was created for application to existing 450 K datasets to expand cell-type representation.

## Results

**Deconvolution library development**. To define a novel deconvolution library from 12 purified cell types, we measured DNA methylation and performed rigorous quality assessment and control for all samples. The final reference dataset included the following cell types: neutrophils (Neu, $n = 6$), eosinophils (Eos, $n = 4$), basophils (Bas, $n = 6$), monocytes (Mono, $n = 5$), B naïve cells (Bnv, $n = 4$), B memory cells (Bmem, $n = 6$), T-helper CD4 + naïve cells (CD4nv, $n = 5$), T-helper CD4 + memory cells (CD4mem, $n = 4$), T regulatory cells (Treg, $n = 3$), T-cytotoxic CD8 + naïve cells (CD8nv, $n = 5$), T-cytotoxic memory CD8 + cells (CD8mem, $n = 4$), and natural killer cells (NK, $n = 4$). The estimated purity of reference samples is based on commonly accepted CD marker definitions (Supplementary Table 1). The cells represented a wide variety of adult donors, including both sexes and different genetic ancestries, a more granular demographic information is included in Supplementary Table 2. The mean purity obtained from the flow cytometry confirmation step (after antibody-linked magnetic bead sorting) was 93% (range 85–99%), with the lowest purity observed for the CD8mem samples (85%). We first used the minfi pickCompProbes function to select an automatic library and estimated cell-type proportions using this library with methylation data from purified cells. This library represents the expected average signal of some extreme hypo and hypermethylated markers per cell type. Pure samples would approximate the average signal of the specific cell type. However, when a sample is contaminated, the signal from another cell type(s) will differ from the average and indicate the potential contaminant's proportion. This procedure for reverse cell-type estimation is denoted as "DNA methylation purity." We previously used this technique to corroborate cell identity and estimate potential residual cross-contamination during the flow cytometry procedures[24]. The mean DNA methylation purity for these samples was 97.6% (range: 85.7–100%), where a target cell-type purity of 85% was required to include the sample in the dataset for library construction. Of the cell subtypes included, CD4nv had the highest estimated cell purity (range: 95.2–100%, median DNA methylation purity: 100%, interquartile range-IQR: 96.5–100%), and CD8mem cells had the lowest DNA methylation purity (range: 85.7–98.5%, median DNA methylation purity: 93%, IQR: 88.3–97.8%). The remaining cell types had median DNA methylation purity that ranged between 97.5 and 99.4%, IQR: 97.3–99.4%, (Supplementary Fig. 1). Potential genetic sources of variability were assessed, including known SNPs tracing genetic ancestry (Supplementary Fig. 2). We excluded probes potentially tracking to polymorphisms, cross-reactive areas, or CpHs, probes tracking to sex chromosomes, and those whose signal intensities were equal or below to the background probes (see Methods for details). After filtering, 675,992 high-quality probes were retained for analysis and deconvolution library construction. The first 20 principal components showed that the main sources of variability

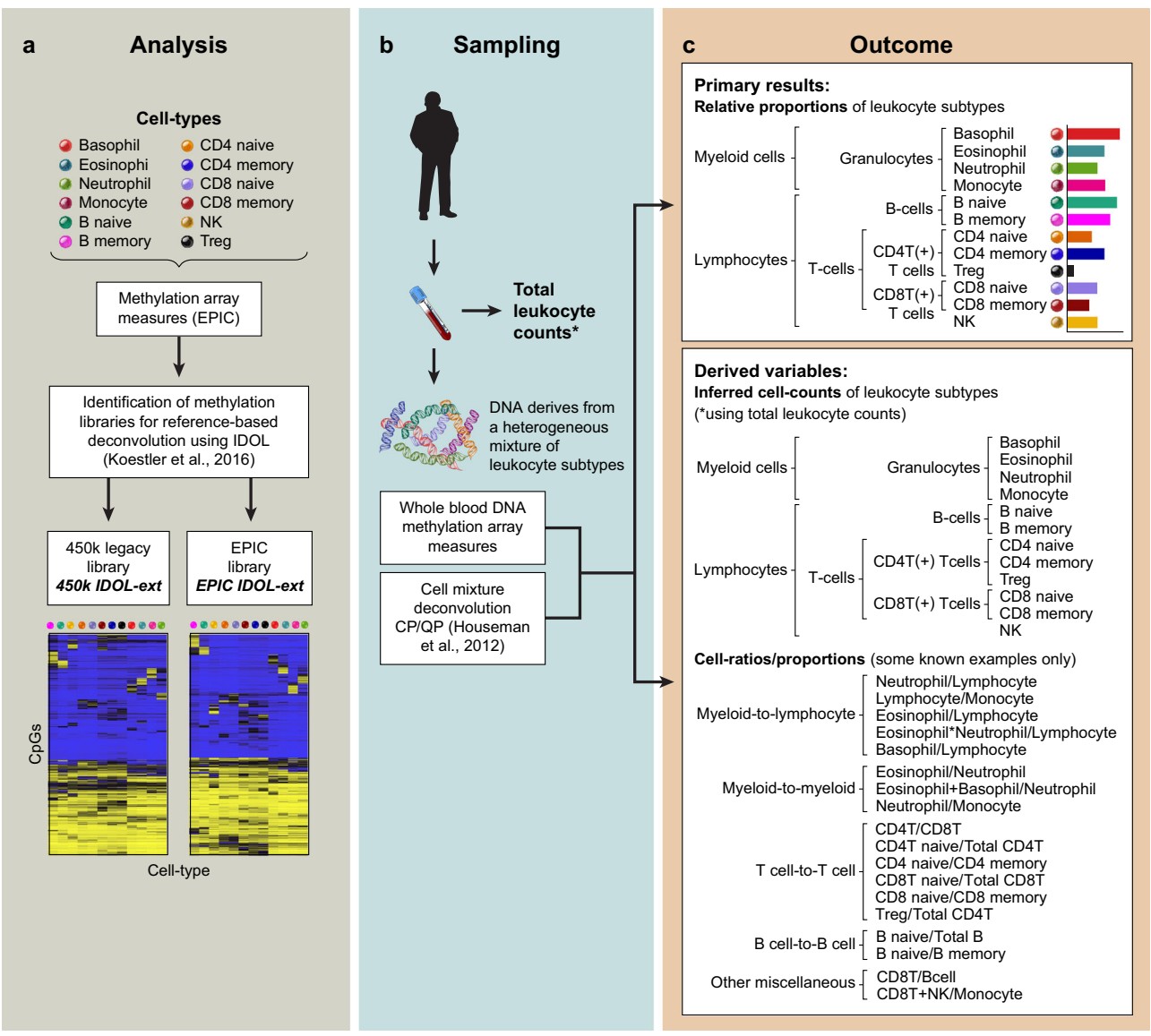

**Fig. 1 Extended library development, samples intended analysis and expected and optional measures from the library application. a** Twelve cell types were acquired commercially, their DNA was isolated, and DNA methylation was measured using the Illumina HumanMethylationEPIC (EPIC) microarray. Using artificial mixtures as ground truth, two libraries were identified using an iterative process named IDOL (IDentifying Optimal Libraries, Koestler, et al. 2016). The two extended (ext) libraries were designed for microarray data derived from the EPIC array (EPIC IDOL-ext) or legacy data derived from the previous Illumina-HumanMethylation450k array (450k IDOL-ext). **b** Samples with variable amounts of leukocytes are arrayed using any of the two validated microarray technologies. Using the appropriate library for the microarray, a cell mixture deconvolution is performed using the constrained projection/quadratic programming (CP/QP, Houseman, et al. 2012). *Optionally, leukocyte counts can be collected for downstream analyses. **c** The primary results of the deconvolution are the 12 cell types of the library. These results could be aggregated at different levels for different hypotheses. Two sets of derived results are possible: (1) if total leukocyte counts are available cell-type-specific counts may be inferred, or (2) cell ratios and proportions are used to evaluate immune-cell shifting between the different cell-type subpopulations (only a few examples illustrated here).

corresponded to the cell-type identity and the slide beadchip, not to sex, age, or other phenotype variables (Supplementary Fig. 3). Due to the distribution of the 12 cell types across 26 different slide chips, some residual experimental variance cannot be corrected in the modeling. As such, we strived for the highest quality data to eliminate most of the additional technical variability in the experiment.

We next applied the IDOL algorithm to the 56 samples collected across all 12 cell types using the 675,992 high-quality CpGs to select the optimal library for accurate deconvolution of these 12 cell types. To test our library's performance, we used artificial mixtures ($n = 24$) of DNA from purified cell types representing varying proportions of the 12 cell types in the IDOL

analysis and measured DNA methylation in these samples (Fig. 2, Supplementary Table 3). Using a discovery selection pool totaling 3535 CpGs representing candidate markers of differentially methylated CpGs across the interrogated cell types, we implemented the IDOL algorithm with 500 iterations to select a set of libraries ranging in size from 250 to 3000 CpGs, in increments of 50 CpGs (detailed in the methods). The coefficient of determination ($R^2$) and the root mean square error (RMSE) were calculated based on a comparison of deconvolution estimates obtained using each library versus the known proportions of the 12 cell types in the artificial mixture samples (Supplementary Table 4). The optimal library, EPIC IDOL-Ext, consisted of 1200 CpGs and was observed to have an average $R^2$ of 1 across all cell types and the

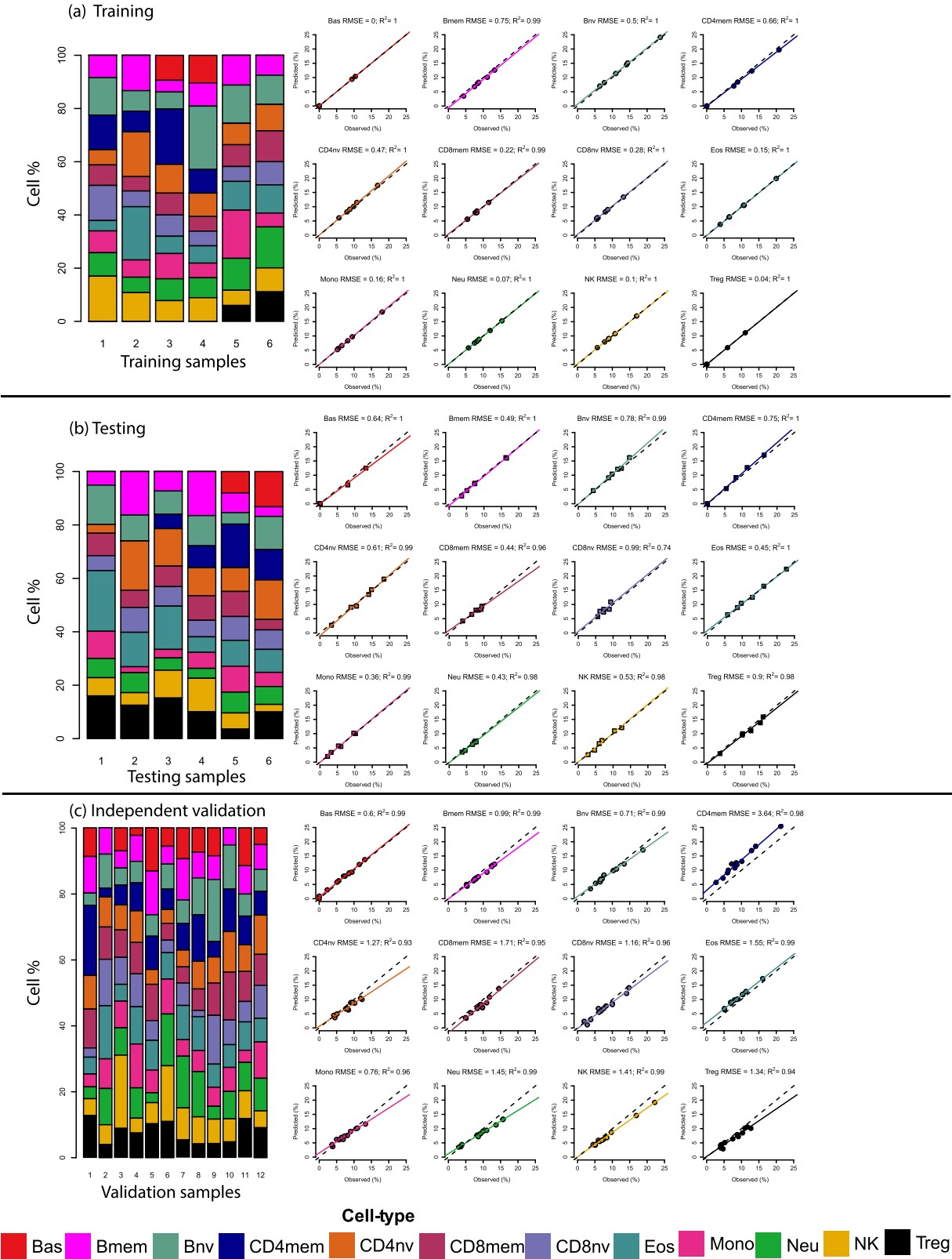

lowest average RMSE (0.226) among the interrogated library sizes (Fig. 2a). The testing dataset showed high R² and low RMSE across the different interrogated cell types (Fig. 2b). The library was further validated using a set of 12 artificial mixtures obtained from independently isolated samples (purity > 90%) not used in the library construction (Fig. 2c), with a high coefficient of determination and low RMSE. We observed some slight increase in the error estimations for CD4mem (RMSE = 3.64) with compensatory decreases in the Treg, CD4nv, and CD8mem compartments.

Probes in the EPIC IDOL-Ext library tracked mostly to open sea regions of low CpG density (76%), followed by CpG island shore regions (14%) (see Table 1), and were enriched for open sea regions (OR: 2.19 95% CI:1.93–2.49), and depleted for probes in

**Fig. 2 Comparison of estimate cell proportions using constrained projection/quadratic programming (CP/QP) versus the reconstructed (true) DNA fraction in the artificial DNA mixtures using the EPIC IDOL-Ext method. a** Cell-specific DNA proportions per sample included in the training set, $R^2$ and Root Mean Square Error-RMSE using the EPIC IDOL-Ext method per cell type. **b** Cell-specific DNA proportions per sample included in the testing set, $R^2$ and RMSE using the EPIC IDOL-Ext method per cell type. **c** Cell-specific DNA proportions per sample included in an independent validation set (using isolated cells not included in the testing or training), $R^2$, and RMSE using the EPIC IDOL-Ext method per cell type. Source data are provided as a Source Data file. Bas Basophils, Eos eosinophils, Neu neutrophils, Bnv B-cells naïve, Bmem B-cells memory, CD4nv helper CD4(+) T-cells naïve, CD4mem helper CD4(+) T-cells memory, Treg CD4(+) T regulatory cells, CD8nv cytotoxic CD8(+) T-cells naïve, CD8mem cytotoxic CD8(+) T-cells memory, Mono monocytes, NK natural killer cells.

**Table 1 Characteristics of the context of the different blood cell-type deconvolution libraries.**

|  | EPIC IDOL-Ext n = 1200 N (%) | 450k IDOL-Ext n = 1500 N (%) | EPIC IDOL-6 n = 450 N (%) | pickCompProbes n = 1200 N (%) |
|---|---|---|---|---|
| Genomic context |  |  |  |  |
| CpG Island | 50 (4) | 122 (8) | 10 (2) | 62 (5) |
| Shores | 189 (16) | 388 (26) | 63 (14) | 146 (12) |
| Shelves | 95 (8) | 213 (14) | 35 (8) | 90 (8) |
| Open Sea | 866 (72) | 777 (52) | 342 (76) | 902 (75) |
| Functional context |  |  |  |  |
| Promoter | 276 (23) | 558 (37) | 104 (23) | 271 (23) |
| Exon | 88 (7) | 144 (10) | 25 (6) | 100 (8) |
| Intron | 551 (46) | 556 (37) | 201 (45) | 548 (46) |
| Intergenic | 285 (24) | 242 (16) | 120 (27) | 281 (23) |
| Enhancers | 139 (12) | 76 (5) | 70 (16) | 194 (16) |
| DHS | 856 (71) | 1001 (67) | 328 (73) | 868 (72) |
| Open chromatin | 135 (11) | 174 (12) | 43 (10) | 113 (9) |
| TFBS | 166 (14) | 193 (13) | 59 (13) | 147 (12) |
| Contained in 450 K | 459 (38) | 1500 (100) | 149 (33) | 517 (43) |
| Total overlap vs. Ref |  |  |  |  |
| vs. EPIC IDOL-Ext | Ref | 330 (22) | 43 (10) | 147 (12) |
| vs. 450k IDOL-Ext | 330 (28) | Ref | 27 (6) | 89 (7) |
| vs. EPIC IDOL-6 | 43 (4) | 27 (2) | Ref | 57 (5) |
| vs pickCompProbes | 147 (12) | 89 (6) | 57 (13) | Ref |

*DHS* DNase Hypersensitive site, *TFBS* transcription factor binding site, *Ref* reference for comparison in the column, Enhancers Phantom5 enhancers. Genomic context, Enhancers, DHS, and open chromatin information were extracted from the Illumina EPIC annotation file "IlluminaHumanMethylationEPICanno.ilm10b5.hg38". Functional context information was extracted from the UCSC reference genome file "UCSC_hg19_refGene.bed".

CpG Islands and Shores (See Supplementary Table 5 for details). Probes were enriched for DNAse hypersensitivity sites-DHS and Phantom5 enhancers. Notably, the library probes were also highly distinct as only 4% overlapped with the EPIC IDOL-6[19].

As only 459 (38%) of the 1200 probes in the EPIC IDOL-Ext library are common to both the EPIC and 450k array platforms, we developed a library for the legacy IlluminaHumanMethylation450k array platform, repeating the above-described optimization process after constraining the selection pool for the candidate list of CpGs to those only present on the 450k array. The resulting library, 450k IDOL-Ext, contains 1500 probes; the library details are included in Table 1 and Supplementary Table 5. One thousand five hundred contained a lower proportion of open sea (52%) and enhancer-related probes (5%) than the EPIC IDOL-Ext library, although Phantom5-enhancer probes and DHS probes were enriched compared to the total background probes in the microarray. In total, 330 probes were shared between the 450k IDOL-Ext and EPIC IDOL-Ext libraries.

Finally, we compared the EPIC IDOL-Ext, and 450k IDOL-Ext versus the pickCompProbes EPIC library obtained using functions in the minfi Bioconductor package[11]. The pickCompProbes automatic selection method builds the library picking the top 50 most hyper- and hypomethylated CpGs per cell type, totaling 1200 probes (same size as our EPIC IDOL-Ext library), summarized in Table 1 and Supplementary Table 5. The probes overlap from the pickCompProbes library with the EPIC IDOL-

Ext library was only 147 (12%). However, the probes' genomic context distribution was similar between libraries, with similar enrichment of open sea and larger Phantom5-enhancer regions than the EPIC IDOL-Ext. When analyzing the cell type by cell-type estimation performance across all three libraries, there was consistency in monocytes, neutrophils, and NK cells. However, more variability was observed for estimates obtained from the library derived by pickCompProbes. When assessing the deconvolution accuracy of the pickCompProbes library, the RMSE was severely biased for eosinophils and T-cell subtypes. Specifically, CD4 and CD8 naïve T-cell distributions were biased with an underrepresentation of CD8nv (RMSE: 5.81%) and the overestimation of CD4nv (RMSE: 9.25%). The pickCompProbes library also had unreliable results for CD4mem vs. Treg compartments and Eos, Supplementary Fig. 4. In contrast, both the EPIC IDOL-Ext and 450k IDOL-Ext libraries were highly accurate in the training and testing datasets, Fig. 2 and Supplementary Fig. 4. The heatmaps summarizing the markers in the three libraries are shown in Fig. 3. The complete set of markers information is available as Supplementary Data Files 1 (EPIC IDOL-Ext), 2 (450k IDOL-Ext), and 3 (pickCompProbes).

**Libraries validation.** The EPIC IDOL-Ext library was validated using samples with blood cell counts from flow cytometry (FCM) and by using an independent set of artificial mixtures from the Gene Expression Omnibus (GEO) (Fig. 4). In five samples (Fig. 4a,

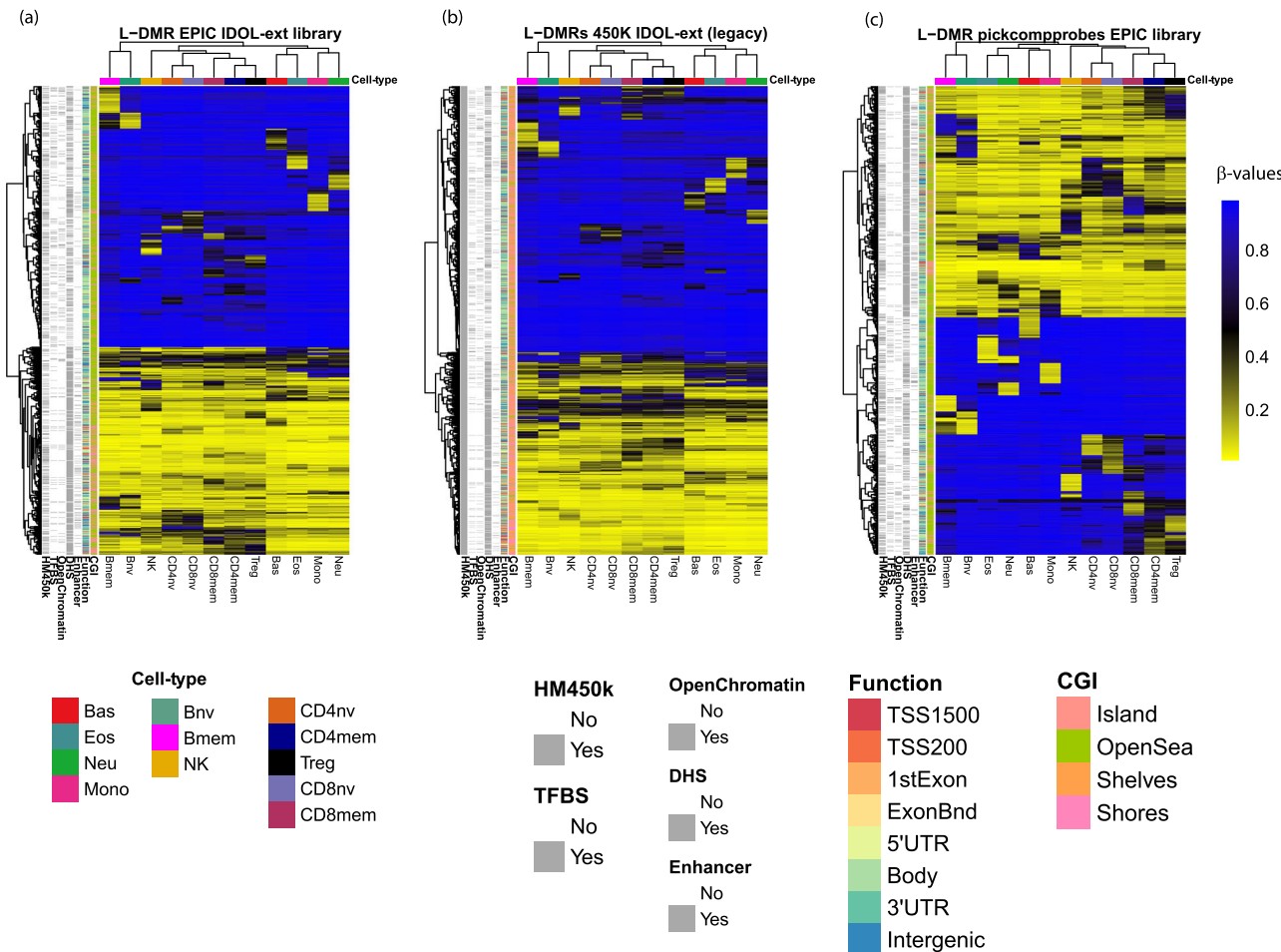

**Fig. 3 Comparison of the selected CpG among the EPIC IDOL-Ext, 450k IDOL-Ext (legacy), and pickComProbes. a** DNA methylation of the 1200 CpG probe EPIC IDOL-Ext library with average methylation for cell-type samples shown in columns as labeled and tracking bars for CpG probes information in the rows, including: present on 450k array (HM450k), tracking to transcription factor binding site (TFBS), DNA hypersensitivity site (DHS), enhancer region from Phantom5 annotation, and genomic context relative to gene (Function) and CpG island (CGI). **b** DNA methylation of the 1500 CpG probe 450k IDOL-Ext (legacy) library and **c** DNA methylation of the 1200 CpG pickComProbes library. The average DNA methylation levels (here as beta values) are represented per each of the 12 cell types. Information about CpG island gene context, Phantom5-enhancer information, DNase Hypersensitivity sites (DHS), Open chromatin, and annotated transcription factor binding sites (TFBS) from ENCODE included in the Illumina annotation file are summarized using the row ribbons. Source data are provided as a Source Data file. Bas Basophils, Eos eosinophils, Neu neutrophils, Bnv B-cells naïve, Bmem B-cells memory, CD4nv helper CD4(+) T-cells naïve, CD4mem helper CD4(+) T-cells memory, Treg CD4(+) T regulatory cells, CD8nv cytotoxic CD8(+) T-cells naïve, CD8mem cytotoxic CD8(+) T-cells memory, Mono monocytes, NK natural killer cells.

GSE110530) from a healthy male subject in his forties, who were followed longitudinally for 400 days, we observed a strong correlation between the deconvolution estimates and FCM measurements, with a maximum root mean square error- RMSE of 2.15% for CD4T cells[19]. Using independent artificial mixtures from the previously published FlowSorted.Blood.EPIC library with six known cell types (Neu, CD8T, CD4T, B cell, Mono, NK; Fig. 4b, GSE110554), the correlation was close to 1, and the maximum RMSE was 3.39 for the CD8T. The 450k IDOL-Ext library was also validated using the GSE77797 dataset with six anonymous adult volunteers [5 males, epigenetic age mean (sd): 34.1(11.4) years] with FCM information (Fig. 4c) and six artificial mixtures with six known cell types (Neu, CD8T, CD4T, B cell, Mono, NK). The information from this dataset was used in the original IDOL manuscript (Fig. 4d)[12]. Of note, the coefficient of determination was lower for monocytes (0.19) and NK (0.66), Fig. 4a, c, respectively, using the EPIC IDOL-Ext library. This decrease is driven by the narrower range of distribution of these cells in the FCM samples. Consequently, any minor deviation in the estimation is magnified even with the high precision reflected by the RMSE.

As a proof-of-concept, we explored how the cell proportion estimates from the EPIC IDOL-Ext library could recapitulate cell counts using FCM cell count information from the GSE112618 dataset (Supplementary Fig. 5)[19]. This dataset included six subjects [five males: mean (sd) age 42.8 (10.6) years] with counts for nine cell types (Bas, B cell, CD4T, CD8T, Eos, Neu, Mono, NK, and Treg). Additional validation sets were analyzed, including glioma patient blood and the cytometric information for the Reinius dataset[17]. A group of 76 glioma patients (GSE180683) was used to evaluate the information from T-cell subsets. The patients were 51% male with a mean (sd) epigenetic age (Horvath) of 54.7 (14.1) years. For this study, cytometric information was obtained using a two-stage characterization of T-cell subtypes. Briefly, T cells were measured and separated into CD4 + and CD8 +. Next, using a second tube, samples were characterized as naïve or memory. Although samples were measured from the same subjects, the proportions were estimated based on two independent measurements and were mathematically derived based on both experiments' average counts. We observed the second-largest difference for the CD8mem with an

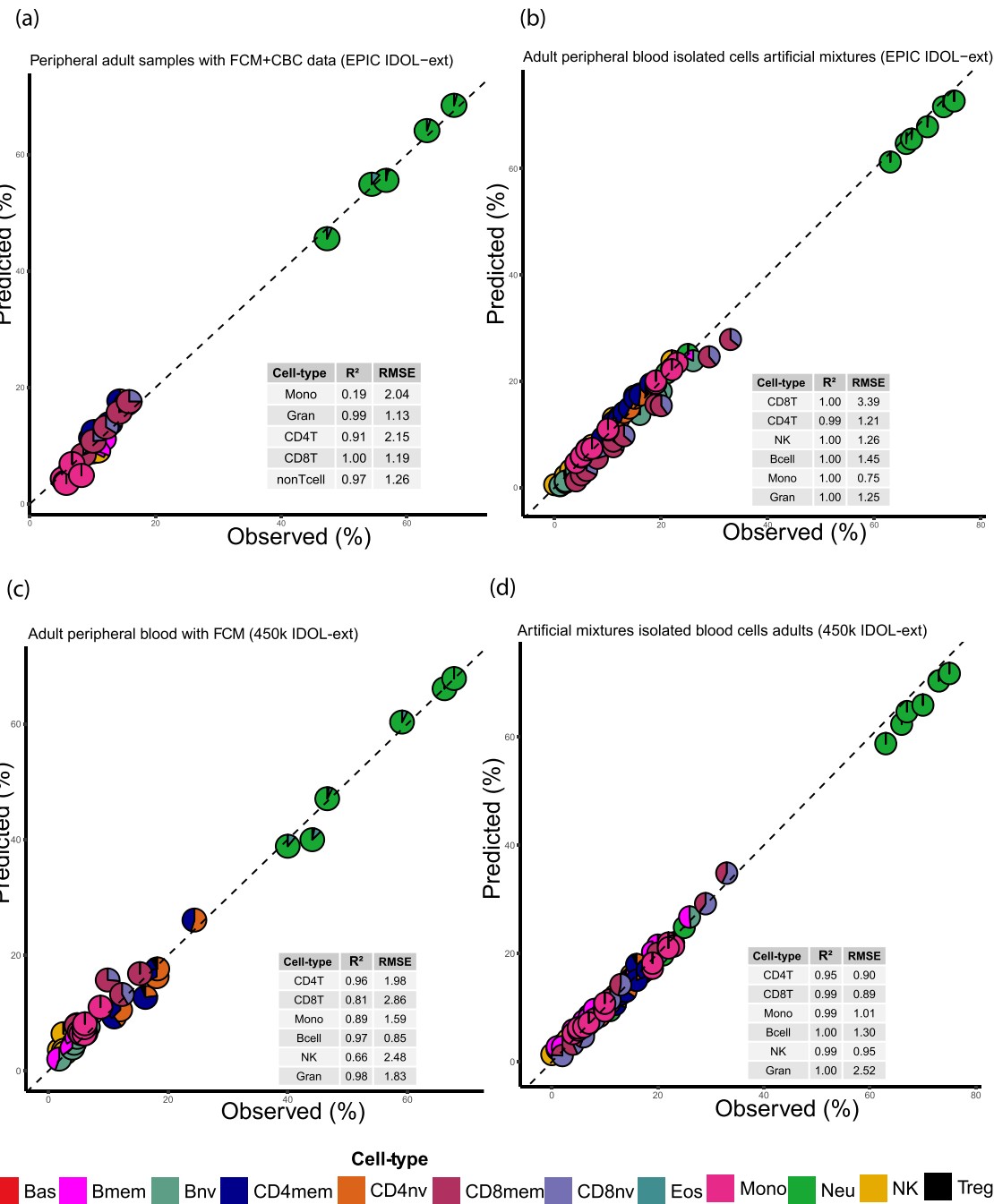

**Fig. 4 Validation of the library using flow cytometry-FCM and independent artificial mixtures for adult peripheral blood. a** and **b** Vaalidation of the EPIC IDOL-Ext. **c** and **d** Validation of the 450k IDOL-Ext. The area of each pieplot corresponds to the estimated proportion of the cell types within each group. Gran (Granulocytes) corresponds to the sum of Neu-neutrophils, Eos-eosinophils, and Bas-basophils. CD4T corresponds to the sum of CD4 + T cells naïve-CD4nv, memory-CD4mem and Treg. CD8T corresponds to the sum of CD8 + T-cells naïve-CD8nv and memory-CD8mem. B cell to the sum of the naïve-Bnv and memory-Bmem. No basophils were detected in the samples illustrated in **a**. Additionally, in **a**, "nonTcell" (Bnv, Bmem, and NK) were not experimentally measured. Source data are provided as a Source Data file.

RSME of 4.34% and the largest with the unmeasured remainder cells (RMSE of 12.6%) (Supplementary Fig. 6a). The 450k IDOL-Ext was also validated using the GSE35069 datasets (Reinius) with FCM information (Supplementary Fig. 6b). The Reinius dataset was composed of six male subjects in their thirties. FCM information was obtained for whole blood, peripheral blood mononuclear cells, and granulocytes for six cell types (granulocytes, CD8T, CD4T, Mono, NK, and B cell). The FCM information is abstracted from the supplementary material in the original manuscript by Reinius et al. We observed

considerable variation (~10%) for the granulocytes, monocytes, and CD4T cells in this dataset[17]. Differences were more prominent for the PBMC samples' observed values and in two of the six WBC samples, particularly for the granulocytes reported.

Finally, we explored whether these libraries could be applied for the deconvolution of umbilical cord blood samples. We used the Jones et al. GSE127824 dataset (450k) with known FCM information for seven cell types (erythroblasts-nucleated red blood cells-nRBC, granulocytes, CD8T, CD4T, Mono, NK, and B

cell)[24]. As erythroblasts were not present in our libraries, these cells are only briefly present in newborns' blood but not at any other age, the fraction was added to the myeloid granulocyte components for comparison. The rationale for adding the erythroblasts to the granulocytes is based on the known biology that establishes the closest lineage to the erythroid-myeloid precursor[25]. This is reinforced by clustering of nucleated red blood cells with basophils and eosinophils using the markers in EPIC IDOL-Ext (Supplementary Fig. 7a). The nRBC fraction showed the most considerable variability in this dataset, with an RMSE of 16.02% (Supplementary Fig. 7b). In contrast, in an experiment of 12 umbilical cord blood artificial mixtures (GSE180970) from healthy anonymous newborns acquired from commercial vendors with only six cell types (granulocytes, CD8T, CD4T, Mono, NK, and B cell), in which the largest RMSE observed for granulocytes was less than 3% (Supplementary Fig. 7c). Of importance, the B cells and T cells clustered with the purified adult naïve cells, while NK, monocytes, and neutrophils clustered perfectly with their adult equivalents.

**Biological interpretation of the libraries' components**. The different libraries' components were examined for how the included genes were related to specific cell types and immune pathways. First, we used the molecular signatures database (MSigDB v. 7.2), curated by the Broad Institute, for gene set enrichment analyses. Most of the pathways were immune-related and are summarized in Supplementary Data File 4. Some specific genes showing differential hypomethylation for specific cell types are outlined in Fig. 5. We also used eForge (https://eforge.altiusinstitute.org/) to identify whether the components of the library were related to particular primed enhancer histone marks (H3K4me1). When analyzing probes specific for the 12 cell types, all subsets were enriched for blood components, but the top significant results were consistent with the specific cell type: E029 Monocytes with monocytes (Supplementary Fig. 8 panel f), E032 B cells with Bnv, and Bmem (panels b and a), E033 T cells with CD4nv, CD4mem, Treg, CD8nv and CD8mem (panels i, h, j, l, and k, respectively) and E046 Natural killers with NK (panel g). There were no specific datasets to compare the granulocytes (Bas, Eos, Neu), where the signal was less straightforward (panels c, d, and e).

**Applications of the libraries**. We next applied the EPIC IDOL-Ext library and the 450k IDOL-Ext libraries to several publicly available datasets from GEO and ArrayExpress to identify potential variation in immune-cell proportions in multiple sclerosis, rheumatoid arthritis, breast cancer patients, and COVID-19 infection (Supplementary Table 6). In multiple sclerosis patients, we observed significantly increased basophil and naïve B cell proportions in cases ($n = 13$) compared to controls ($n = 14$) (Wilcoxon rank-sum $P < 0.01$, Supplementary Fig. 9). In rheumatoid arthritis, significant increases in neutrophil and regulatory T-cell proportions and decreases in memory B cell, memory CD4 cell, naïve CD4 cell, memory CD8 cell, naïve CD8 cell, eosinophil, monocyte, and NK cell proportions were observed in case blood samples ($n = 354$) compared to control blood samples ($n = 355$) (Wilcoxon rank-sum $P < 0.01$, Supplementary Fig. 10). Predicted immune-cell proportions in breast cancer patients before and after receiving chemotherapy or a combination of chemo/radiation therapy were explored. Patients receiving radiation therapy only ($n = 74$) showed a significant relative increase of the neutrophil proportion and a mirror decrease of several lymphoid lineages (memory B cell, naïve B cell, naïve CD4 cell, and NK) after treatment (Wilcoxon rank-sum $P < 0.01$, Supplementary Fig. 11a). Patients receiving

radiation therapy and chemotherapy ($n = 70$) exhibited significant increases in eosinophil, monocyte, and regulatory T-cell proportions and decreases of memory B cells after treatment (Wilcoxon rank-sum $P < 0.01$, Supplementary Fig. 11b). Finally, we evaluated the changes in immune-cell proportions in six COVID-19 patients with and without remission versus six healthy controls in Supplementary Fig. 12. Because of the limited sample size, no statistically significant differences were observed, but, as expected, the median of neutrophils was higher in patients versus controls. In contrast, all the median lymphocyte subpopulations were lower in the infected patient than those who recovered from the disease. The median monocytes were lower in those that remitted.

We investigated blood methylation data associated with subject-to-subject variation in non-pathological conditions using data from monozygotic versus dizygotic twins and subjects at different ages (Supplementary Table 6). To characterize immune cell variation between twins, predicted immune cell proportions in monozygotic ($n = 852$) and dizygotic twins (n = 612) were estimated. Significant differences in all the immune cells were observed in monozygotic twins and dizygotic twins (Paired $t$-test $P < 0.01$). More considerable differences between dizygotic twins and monozygotic twins were seen in memory B cell, naïve B cell, memory CD4 cell, naïve CD4 cell, memory CD8 cell, naïve CD8 cell, eosinophil, monocyte, and NK (Wilcoxon rank-sum $P < 0.01$, Supplementary Fig. 13). Next, the impacts of aging on predicted immune cell proportions in blood samples from newborn to nonagenarian ($n = 2504$) were investigated. Extensive literature has explored the changes in immune proportions with aging[26,27], but it has not been systematically explored using DNA methylation[28–30]. Several subpopulation cell ratios across different ages were calculated (Supplementary Fig. 14). The complete granular data and trajectories of the different cell subpopulations are represented in Supplementary Fig. 15. Longitudinal changes of predicted immune cell proportions within five years after birth in human blood leukocytes from 10 healthy girls are shown in Supplementary Fig. 16.

We included some sensitivity analyses to compare the 450k IDOL-Ext vs. the EPIC IDOL-Ext libraries when using EPIC or a combination of EPIC and 450k legacy arrays. As shown in Supplementary Fig. 4, both libraries provide an accurate estimation for the 12 cell types. There are minor variations in Bmem and Bnv estimations, with more precise estimates using the EPIC IDOL-Ext (absolute mean difference 0.08%-EPIC vs. 0.48%–450k, and 0.23-EPIC% vs. 0.52%–450k, respectively), and CD4mem and Treg with more precise estimations using the 450k IDOL-Ext (absolute mean difference 0.62%-EPIC vs. 0.07%–450k, and 0.6-EPIC% vs. 0.34%–450k, respectively), see Supplementary Fig. 17 for additional information. Results using other statistical procedures for cell deconvolution, CIBERSORT, and robust partial correlations, were compared and are illustrated in Supplementary Fig. 18. Results were generally similar using the different methods, though slightly lower accuracy was observed for CIBERSORT compared to CP/QP or robust partial correlations.

## Discussion

We established compact and reliable libraries to deconvolve the proportions of 12 different cell types in peripheral blood, including closely related cell types such as T-cell subtypes and various types of granulocytes. Importantly, the libraries' derived variables offer a detailed immune profile from peripheral blood, providing more than 56 cell-type and ratio components. Our libraries are designed both for current (EPIC) and legacy (450k) DNA methylation measurement platforms. The Gene Expression

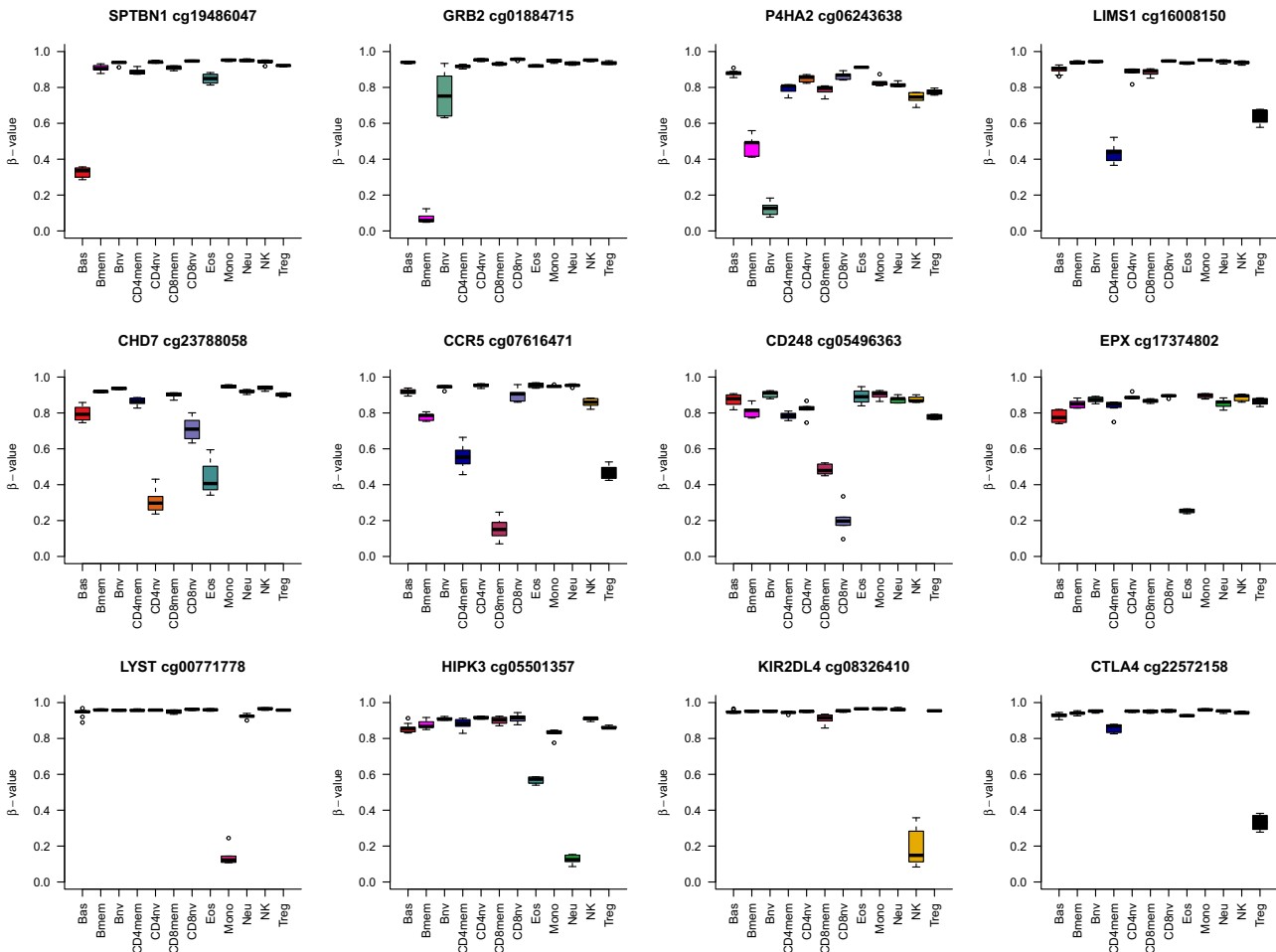

**Fig. 5 Examples of differentially methylated sites demarcating the different cell lineages.** The boxplots include the following information: (1) The box shows the interquartile range (IQR), (2) the whiskers show the inner fences (1.5 × IQR out of the box), (3) the bolded line shows the median of the data. The color inside the box corresponds to the cell type. Source data are provided as a Source Data file. Bas Basophils, Eos eosinophils, Neu neutrophils, Bnv B-cells naïve, Bmem B-cells memory, CD4nv helper CD4(+) T-cells naïve, CD4mem helper CD4(+) T-cells memory, Treg CD4(+) T regulatory cells, CD8nv cytotoxic CD8(+) T-cells naïve, CD8mem cytotoxic CD8(+) T-cells memory, Mono monocytes, NK natural killer cells.

Omnibus repository, as of July 2021, includes publicly available data from ~148,325 samples for both platforms[31]. Our libraries have high accuracy and minimal bias compared to flow cytometry, considered the "gold standard" data (Fig. 4). For most cell types, that accuracy represents a difference of less than 3% in the estimations. However, more considerable differences (>4%) are observed in some datasets for the most abundant cell types (neutrophils and total CD4T, Supplementary Figs. 6 and 7) and for CD8Tmem (Supplementary Fig. 6). This error results from limited purity in this cell component and possibly differences in gating processes during the count splitting[32]. For the CD8mem compartment, it is also possible that unaccounted residual heterogeneity is present, TEMRA (T effector memory cell expressing CD45RA isoform), effector memory, central memory, NKT cells, and other less characterized CD8Tmem subclasses[33]. The new libraries will be important where specific memory and naïve cell compartments are interrogated for disease, pathogenesis, or exposure. Both EPIC IDOL-Ext and 450k IDOL-Ext are accurate, with the former requiring fewer probes for the estimation. In complex problems, such as EWAS using EPIC and 450k technologies simultaneously, we recommend using the more straightforward approach for the data using the 450k IDOL-Ext given the minimal variation observed (<1%) in the estimates. We advise the researchers to evaluate their data quality before using any deconvolution approach to avoid potential technical biases

(e.g., examining the library probe call rate and possible batch effects in the data).

We consistently observed biologically relevant genes and groups of genes in related pathways for the different cell types, such as the *SPTBN1*, which is more expressed in basophils[34]; *GRB2*, involved in Bmem formation[35,36]; *P4HA2*, related to immature B cells[37]; *CHD7*, associated with thymic function[38]; *CCR5*, related to CD8mem recruitment and trafficking[39]; *CD248* a CD8 regulator and marker of naïve states[40]; *EPX*, the eosinophil peroxidase marker of eosinophils[41]; *LYST* a lysosome maturation in monocytes[42]; or *KIR2DL4* a natural killer cell receptor[43]; and the *CTLA4* an essential marker in Treg tumor suppression[44].

Our EPIC IDOL-Ext and 450k IDOL-Ext libraries provide enhanced features for researchers using DNA methylation data derived from blood. When the research question requires precise resolution, the newly extended libraries provide that extra information, with a minimal trade-off in precision compared with the earlier EPIC IDOL-6 library. Previous attempts to generate extended libraries for blood deconvolution encountered several problems. The discrimination of scarcer cells has been problematic, leading to inconsistent estimates; even when candidate references were available, this library has largely overcome those challenges. Some markers delineating particular cell types are dependent on sex chromosome dosage (e.g., *FOXP3* located in the X chromosome for Tregs). To avoid that potential problem here,

we only used autosomal probes. Distinguishing hierarchically close cell types such as CD4mem and Tregs is challenging as the pool of potential specific markers is limited. These markers may even show some dependency when transitioning from one state to another. For example, *CD248* is mostly unmethylated in CD8nv, but some cells in the CD8mem compartment are not completely methylated at *CD248*, while it is completely methylated and repressed in other cell types. In such case, using multiple markers attenuates the effect of using single markers to define cell identity. Finally, the statistical process may favor markers that do not necessarily reflect biologically relevant genes when using libraries derived from an automatic selection. For example, four (EPIC) and five (450k) *EPX* probes were selected in the extended libraries, in contrast to none by the minfi pick-CompProbes procedure. *EPX* is the gene containing the information for the eosinophil peroxidase protein, which is essential for eosinophil function and solely expressed in this blood cell type[41,45].

Including six additional cell types and, in particular, the naïve compartments is a significant step towards a universal deconvolution library for application to cord or adult blood data. Particularly intriguing for umbilical cord blood, nucleated red blood cells interfered with more precise estimations than available FCM information. Yet artificial mixtures showed the library capturing leukocyte components accurately. As expected, most (if not all) of the B and T cells were naïve cells as memory populations are derived after antigen exposures, most commonly after birth.

Another advance of this library is the most accurate eosinophil predictions compared to previous libraries and now basophil estimations. These cells are involved in multiple pathologies, including allergies and asthma, and through these libraries, they can now be estimated. As with previous libraries, some residual confounding from unaccounted cell types may be observed. In particular, we observed inflation in several datasets of the basophil estimation. Based on single-cell murine data, one hypothesis for this estimate inflation is common progenitors' relation with triple differentiation to basophil/mast cells and erythroid cells[46,47]. According to those potential common markers, it is possible the basophil signal could interfere with the signal derived from circulating immature nucleated erythroid cells. At least in umbilical cord blood samples, the signal obtained from basophils is highly correlated with the predicted fraction of nucleated red blood cells. As such, a careful assessment of any high basophil signal is warranted, as it could be driven by interference with nucleated red blood cells in newborns and potentially some pathological or adaptative conditions at other ages. Thus, careful interpretation based on those estimates alone is prudent as they may signal residual cell signals of other cell types not included in the library.

We applied this library to several public datasets, comparing cases and controls for several pathologies. Our results are concordant with known pathophysiology and immune responses for these diseases: increased proportions of abnormal naive B cell populations in the blood in multiple sclerosis patients[48], pancytopenias with a predominance of neutrophils in radiotherapy plus chemotherapy in breast cancer[49], predominancies of neutrophilia, and lymphopenia in COVID-19 patients that did not show remission[50]. However, the limited sample sizes of these data preclude more generalizable conclusions for most of the findings. We will only discuss the most extensive dataset available of rheumatoid arthritis patients vs. controls. This dataset, published by Liu et al[51]., has explored how some immune cell components (monocytes) are associated with specific DNA methylation changes in the cases who were not receiving any treatment at the time of blood collection. We observed a reduction in monocyte proportions in rheumatoid arthritis patients compared to controls (P-value = 1.1 E-5). Paradoxically in our analysis, we found a non-statistically significant increase in Tregs in cases compared to controls (P-value = 0.06), which are related to disease activity or response to medications in other studies[52]. These differences are essential to understand the activity of the disease and potentially for the classification of the heterogeneous subjects when compared to healthy controls. It is difficult to predict the extent of discoveries made possible with the extended capability for immune profiling we provide. Our method gives higher resolution than clinical approaches allows for more scalable, efficient, standardized immune profiling compared with flow cytometry, and has the added benefit of applying to archival specimens. Although we have only begun to demonstrate the potential of our approach, we do not doubt that the research community will leverage this method to provide new insights into the relation of immune profiles in human health and disease in exciting and meaningful ways that impact public health and translational medicine.

In summary, the new reference libraries greatly enhance the detail of immune cell profiling with DNA methylation. These enhanced libraries can be applied directly for immune profiling and adjustment of cell-type proportions in EWAS. Future work includes validating the use of this library in methylation data from children and umbilical cord blood and expanding the libraries to additional cell subsets critical for some pathologies (e.g., dendritic cells). The increased detail in describing leukocyte subtypes using the extended libraries will be crucial for controlling for aging-related DNA methylation changes; age-associated changes in specific subpopulations of memory T and B lymphocytes are unaccounted for in previously established libraries. Indeed, the IDOL-Ext library applications developed here greatly enhance the detail of immune characterization for existing immune profiles data and open opportunities for future studies using DNA methylation data derived from human blood samples.

## Methods

This work extended the available six-cell reference library using twelve cell subtypes for deconvolution of blood cell proportions using the EPIC array, as well as a legacy library for the 450k platform. Using cytometric and magnetic-sorted, flow confirmed neutrophils, eosinophils, basophils, B cells (naïve and memory), monocytes, NK cells, CD4 + T cells (naïve, memory and T regulatory cells), and CD8 + T cells (naïve and memory), DNA methylation was measured with the 850 K/EPIC DNA methylation array. We applied the IDOL method to identify optimal Leukocyte -Differentially Methylated Regions (L-DMR) libraries using a testing set of six artificial mixtures containing the 12 cell types. Artificial mixtures (also referred to as reconstructions) consist of DNA from purified isolated cell types, representing mock blood samples of known, predefined cell proportions. Six additional testing artificial mixtures were employed to corroborate the performance, and 12 independent artificial mixtures derived from a set of 12 isolated cell types not utilized in the training or testing, were used to validate the results. We compared the performance of cell estimates obtained applying our previously developed six model cell type (available in Bioconductor as "FlowSorted. Blood.EPIC") and optimized an additional L-DMR IDOL library limiting the probes to those available in the older 450 K array, and again compared the performance using the training, testing, and validation datasets.

The DNA used to generate mixtures were derived from four MACS-isolated and FACS-verified purity cell subtypes from the myeloid lineage [neutrophils (Neu), eosinophils (Eos), basophils (Bas), and monocytes (Mono)], and eight MACS-isolated and FACS-verified purity cell subtypes from the lymphoid lineage [B lymphocytes naïve (Bnv), B lymphocytes memory (Bmem), T-helper lymphocytes naïve (CD4nv), T-helper lymphocytes memory (CD4mem), T regulatory cells (Treg), T-cytotoxic lymphocytes naïve (CD8nv), T-cytotoxic lymphocytes memory (CD8mem), and natural killer lymphocytes (NK) cells] were purchased from AllCells® corporation (Alameda, CA, USA), StemExpress (Folsom, CA), and STEMCELL Technologies (Vancouver, BC, Canada). Cells were isolated from 41 males and 15 females, all anonymous healthy donors. The donors had a mean age of 32.2 years (sd = 12.2, range 19–58 years) and an average weight of 85.1 kg (range 57–136 Kg). Donors identified themselves from multiple ethnicities, including mixed ethnicities, and were categorized broadly into four groups (African-Americans, East-Asian, Indo-European, multiple/admixed). They were negative for Human Immunodeficiency Virus-HIV, Hepatitis B Virus-HBV, and Hepatitis C Virus-HCV. Women were not pregnant at the time of sample collection, and samples were collected from donors with no history of heart, lung, kidney disease, asthma, blood disorders, autoimmune disorders, cancer, or diabetes. All donors

provided written informed consent before donation. The data discussed in this publication have been deposited in NCBI's Gene Expression Omnibus (Salas et al., 2021) and are accessible through GEO Series accession number GSE167998. Isolation protocols are available through the commercial websites of AllCells, StemExpress, and STEMCELL Technologies. In brief, cells were selected using immunomagnetic labeling through the vendors' specific protocols (see, Supplementary Table 1 for details). Recovered cells were confirmed using flow-sorting. Twenty-four artificial mixtures were determined by randomly generating proportions from a twelve-component Dirichlet distribution. Each mixture of 1.2 μg total DNA was generated from isolated cell DNA using the proportions in Supplementary Table 3. The isolated cell DNA and those of the artificial mixtures were bisulfite converted and processed according to the Illumina protocols at the Vincent J. Coates Genomics Sequencing Laboratory at UC Berkeley, Avera, or Diagenode. Samples were randomized prior to loading onto microarray chips. The EPIC methylation array raw idat files were pre-processed using minfi, EnMIX, and SeSaMe for quality control using R v.4.0.2 and 4.1.0[53–55]. To assess data quality, we used an out-of-band detection P-value of 0.05, three standard deviations of the mean bisulfite conversion control probe fluorescence signal intensity, and a minimum of three beads per probe. To ascertain the highest purity of the samples included in our library, in addition to the information obtained through the FCM confirmation, we projected back the proportions, thus "purity," of the cells, using Jaffe's procedure[11]. The corresponding cell-type proportion was retrieved and designated as "DNA methylation purity." Only samples with DNA methylation purity levels higher than 85% (range 85.7 to 100%) were included in the library. Subsequently, a stringent out-of-band p-detection value (pOOBHA) > 0.05 was applied and set those that could not be distinguished from the background probes as "missing values." Only those probes with complete information for all the samples were selected for the library. No imputation was performed in this context as the signals could be dependent on the specific cell type. Additionally, all non-CpG (referred to as CpH) probes were filtered in light of the minimal variation and all CpH beta values were under 0.08. Finally, as both female and male samples were present, we discarded probes tracking the X and Y chromosomes. According to Zhou et al., those that showed known polymorphisms or cross-reactivity were also excluded[56]. Our set for library discovery included 675,992 complete high-quality probes. The EPIC IDOL-Ext L-DMR library is available as an R library "FlowSorted.BloodExtended.EPIC" please contact the Office of Technology Transfer Technology.Transfer@dartmouth.edu for a free Academic license (for license instructions, please refer to https://github.com/immunomethylomics/FlowSorted.BloodExtended.EPIC). The extended blood deconvolution can be performed using the FlowSorted.Blood.EPIC Bioconductor library and we recommend using the minfi noob background correction for the target dataset. The package contains an RGChannelSet R object processed using SeSaMe in which probes showing channel switching were corrected and SNPs derived from Infinium Type I probes were added, using the total signal intensities, to the control for genetic ancestry. The object is unfiltered and contains 56 samples and the 12 artificial mixtures information on 1,008,711 probes corresponding to 866,091 sites (CpGs and CpHs) using the latest annotation released by Illumina (MethylationEPIC_v-1-0_B5). The reader needs to note that the cells were purified using an immunomagnetic procedure; the name "FlowSorted" is kept for historical reasons and downstream integration with previous minfi pipelines and similar algorithms.

**IDOL algorithm**. For a complete description of the IDOL algorithm, please refer to the original application in Koestler et al.[12]. In brief, the IDOL algorithm utilizes a training dataset (ground truth) consisting of samples with DNA methylation data in which the measured fraction of each of the underlying cell types is known (here corresponding to artificial mixtures with prespecified proportions) as a means to identify a set of probes confirming an optimal reference library for cell mixture deconvolution. A series of t-tests compared the mean CpG-specific methylation between each leukocyte cell type vs. the mean methylation across all the other cell types identified the probes discriminating CpGs (e.g., leukocyte differential methylated regions or L-DMRs) for each specific cell type of the 12 included in this application. CpGs were then rank-ordered using their t-statistics, and the L/2 CpGs with the largest and smallest t-statistic for each K cell type were identified and pooled. Our application set the tuning parameter L to 150 in, consistent with Koestler et al.[12]. A discovery L-DMR library containing the total L*K unique L-DMRs for each cell type forms the IDOL algorithm search space. L-DMR subsets of size <L*K are sequentially selected and examined their prediction accuracy in deconvolving the training dataset samples. The user needs to preselect the library size to balance the accuracy and precision of cell-composition estimates. For the current application of IDOL presented here, we considered libraries ranging from 250 to 3000 CpGs. We initially set increments of 50 CpGs until a size of 1100; then we increased by 100 CpGs until a size of 2000 and tested libraries of 2500 and 3000 CpGs to corroborate the elbow in the error distribution. In the first iteration of the IDOL algorithm, all L*K CpGs constituting the candidate library has an equal probability of being selected to be included in the L-DMR library. We applied the constrained projection/quadratic programming approach[9] to obtain the cell-composition estimates for each sample in the training dataset. These predictions then allow us to calculate the $R^2$ and RMSE (root mean square error) for each cell

type, contrasting the cell estimates versus the known proportion in each sample. Then one-by-one CpGs are removed from the randomly selected DMR library, followed by computation of $R^2$ and RMSE based on cell-composition estimates obtained using the new library. Through this, we can assess the contribution of each CpG in the library in terms of its impact on the accuracy of cell-composition estimates, and then the algorithm modifies the probability of each CpG being selected in subsequent IDOL iterations. The process is repeated at each of the 500 iterations, with the algorithm eventually converging on an "optimal" library for deconvolution (showing the lowest error and highest precision).

**Enrichment analysis**. Gene set enrichment analyses used missMethyl to control for multiple probes bias and the Molecular Signatures Database (MSigDB) version 7.2[57,58]. Hypergeometric tests were used to evaluate enrichment of specific genomic context and functions compared to the background probes used for selecting the candidates in the library. Additionally, eForge was used to evaluate the enrichment of cell-specific probes in the library versus the presence of primed enhancer histone marks (H3K4me1)[59].

**Additional validation sets**. We used five datasets for validation: a set of samples obtained from a longitudinal analysis (GSE110530) with six observations. A set of independent artificial mixtures (GSE110554) derived from six cell types. A set of samples from 72 glioma patients in different treatment stages (GSE180683), including FCM information for T cells CD4 + and CD8 + naïve and memory. A set of 20 umbilical cord blood (GSE68456) with FCM information. A set of 12 umbilical cord blood artificial mixtures including six cell types (GSE180970). A set of samples using the 450k technology (GSE77797) with FCM and artificial mixture information derived from six cell types. Finally, the Reinius et al. GSE35069 dataset using the FCM information for the whole-blood cells, peripheral blood mononuclear cells, and granulocytes as reported in the original manuscript's supplementary material.

**Potential applications**. We identified 12 publicly available datasets from GEO and ArrayExpress that contained DNA methylation data on two different normal health conditions (twins, aging) and four diseases (Supplementary Table 6). The application datasets included whole-blood samples from 426 pairs of monozygotic twins and 306 dizygotic twins, 2504 umbilical cord and peripheral blood samples from newborn to nonagenarian, 13 multiple sclerosis whole-blood case samples and 14 controls, 354 rheumatoid arthritis peripheral blood leukocyte samples, and 355 controls, 144 peripheral blood samples from breast cancer patients before and after receiving chemotherapy and radiation or isolated radiation therapy treatment, and six COVID-19 patients (with 18 samples) and six healthy controls. Illumina Infinium DNA methylation IDAT files were retrieved from GEO and ArrayExpress for application datasets. minfi package from Bioconductor was used to process the data using noob. Four thousand eight hundred and seventy-two samples in total were eventually contained in the application datasets. We estimated the proportion of immune cells in those datasets using the appropriate extended immune cell deconvolution library for the array (EPIC or 450k).

**Reporting summary**. Further information on research design is available in the Nature Research Reporting Summary linked to this article.

## Data availability

The main source dataset (12 cell types, testing, and training artificial mixtures) generated in this study have been deposited in GEO under accession code GSE167998. This Superseries GSE181034 is composed of the following series GSE180683 (glioma samples, validations for T-cell memory subsets), GSE180970 (umbilical cord artificial mixtures), GSE182379 (independent validation 12 cell types artificial mixtures). Additional public datasets analyzed in this manuscript are available in GEO and ArrayExpress with accession numbers: GSE110554, GSE77797, GSE110530, GSE35069, GSE68456, GSE88824, GSE42861, GSE140038, GSE161778, GSE105018, E-MTAB-7069, GSE85042, GSE103189, GSE104778, GSE62219, GSE87571, E-MTAB-7309, GSE87571, GSE12163. Source data are provided with this paper.

## Code availability

The code used for this manuscript has been deposited in Zenodo doi: 10.5281/zenodo.5338513[60]. Instructions to obtain a license for FlowSorted.BloodExtended.EPIC R library are available in GitHub (https://github.com/immunomethylomics/FlowSorted.BloodExtended.EPIC) or through the Technology Transfer Office Technology.Transfer@dartmouth.edu. Free licenses are available solely for Non-Commercial Entities to conduct academic research. All other parties will require a specific license. Most of the licenses are granted less than two business days after the agreement signature.

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

## Acknowledgements

L.A.S. was supported by CDMRP/Department of Defense (W81XWH-20-1-0778) and NIGMS (P20 GM104416-09/8299). D.C.K. was supported by NIH grants P30 CA168524, P20 GM130423, and P20 GM103428. J.K.W. and A.M.M. were supported by NIH grants R01 CA207360 and P50 CA097257. K.T.K. was supported by a 2018 AACR-Johnson & Johnson Lung Cancer Innovation Science (18-90-52-MICH) and NIH grant R01CA253976. B.C.C. was supported by NIH grants R01 CA216265 R01 CA253976, P20 GM130454, and P30 DK117469. Research reported in this publication was supported by an Institutional Development Award (IDeA) from the National Institute of General Medical Sciences of the National Institutes of Health under grant number P20 GM104416. We acknowledge the following Shared Resources facilities at the Norris Cotton Cancer Center at Dartmouth with NCI Cancer Center Support Grant P30 CA023108: Immune Monitoring & Flow Cytometry.

## Author contributions

L.A.S., D.C.K., J.K.W., K.T.K., and B.C.C. conceived the project and designed the experiments. L.A.S. and Z.Z. performed the bioinformatic quality control and analyzed the data. H.M.H. and R.A.B. carried out the wet-lab work, including the spiking of the artificial mixtures used in this experiment (R.A.B.). D.C.K. and A.M.M. supported the statistical analysis. L.A.S. wrote the manuscript with input from all the co-authors. The final version of the manuscript has been reviewed and approved by all the authors.

## Competing interests

J.K.W. and K.T.K. are founders of Cellintec, which had no role in this research. A U.S. Provisional Patent Application entitled: ENHANCED DNA METHYLATION LIBRARY FOR DECONVOLUTING PERIPHERAL BLOOD was filed on February 12, 2021 by Dartmouth College for the IDOL deconvolution libraries invention included in this manuscript. Inventors: Christensen, BC, Salas LA, Kelsey KT, Wiencke JW, and Koestler DC. Application number: 63/148,695. The remaining authors declare no competing interests.
