## [Transparent Peer Review File · Nature Communications]

Reviewer comments, first version:

Reviewer #1 (Remarks to the Author: Overall significance):

The authors present a new method for cell type deconvolution of DNA methylation data, greatly expanding the number of cell types from previous algorithms, including Houseman and other methods.

This paper represents a significant advance. The methodology is robust and the results are convincing. This paper and method promises broad application in the DNA methylation community, as virtually every human blood DNA methylation array study is likely to benefit from this software. Work and conclusions are original and solid. Previous work is appropriately credited, and appropriate context is provided.

Reviewer #1 (Remarks to the Author: Impact):

Paper is likely to be highly cited and influence the entire EWAS field.

Reviewer #1 (Remarks to the Author: Strength of the claims):

Work presented is convincing. Some clarifications indicated here:

Main comments:

Line 144: a lot of different consortium data includes both EPIC and 450k. Is there an approach here that can be applied on all the data. What about a common analysis/IDOL that includes both arrays? Is there a solution for this problem (could be valuable in some contexts)? Is this possible?

Line 240: "Our libraries have high accuracy and minimal bias when compared to flow cytometry gold standard data." Please reference some numbers/estimates here. Progress is shown and this method represents a significant advance but no method is perfect. Please discuss pros and cons.

Line 248: Main comment: Results for specific genes are interesting but given that many of these DNA methylation sites are located in open sea regions and enhancers, a chromatin based analysis approach that does not depend on genes would be appropriate. Why not use eFORGE/eFORGE-TF or another motif or chromatin analysis method to validate independently that these are tissue-specific elements (ideally using an enhancer mark e.g. H3K4me1)? An example of such an analysis is appended as a pdf. The MSigDB analysis is interesting but paper would greatly benefit from an agnostic, chromatin-based method given the amount of enhancer-located sites.

Line 262: Main comment: In what way does the IDOL method compare with other methods, for example robust partial correlations? What are the advantages of this iterative method? Discuss

Minor comments:

line 26 retrospective, do you mean "previous"?

line 31 "Immune system" or "epigenetics of the immune system"?

throughout the ms: continued use of the word library could potentially be confusing. As the authors will know, library is commonly used in experimental genomics with a slightly different angle and of course this is purely computational so context is different. There is a precedent in previous pipelines but potentially worth clarifying at the beginning of the text.

line 80: Explain how you go from 12 to 19

line 106: Why is this lower number in the range (85.7) not included/mentioned in the results discussed immediately afterwards (we go from mean to median, maybe could be clarified and focus on just one measure)?

line 117: "slide beadchip" as a remainder of technical variation. I assume this is later removed. Please discuss/mention this

line 133: I am familiar with CpG Island shores but still not sure if north or south component has much relevance here, please explain and if not clear suggest focusing just on describing "CpG Island shores" as north and south might be confusing to non-experts

Line 150: Re genomic context, not clear if the probe distribution by genomic context for IDOL libraries is any different from the array from which the data was selected.. comparison to background probes on the array and some statistical test could potentially be enlightening in this respect

line 168: Was the flow cytometry done by this study or by a previously published study? Please clarify

Line 187: Why was the component of nucleated red blood cells added to the myeloid granulocyte component specifically?

Line 216: A lot of proportional findings for different diseases are reported.. can one be explained in depth? Suggest to focus on at least one and describe biological interpretation in depth

Figure 3: As the authors know, DHSs are open chromatin, so is "open chromatin" used for ATAC-seq data? Discuss

Figure 4A: Why R2 0.16 for monocytes?

Figure 4C: Discuss R2 0.65 for NK cells.

Line 227: change in immune proportions with age. Can part of the literature on this be referenced?

Line 239: ">100,000 samples" Where is this number from?

Line 256: Genes or probes?

Line 264/265: I'm not sure what EPX absence proves maybe this should be explained further or contextualized?

Line 268: than or from?

Line 273 seems like a comma is missing

line 274: can or "they can"?

Line 276: "Thus careful interpretation based on those estimates alone is prudent." Please clarify

Line 282: "a careful assessment of any high basophil signal is warranted" Could the tool include a flag/warning for these high basophil signals as default?

Line 289 "controlling" or "controlling for"

Line 289: "effects", do you mean "DNA methylation changes"?

line 290: "for" or "for in"

Line 292: "to" or "for"

Line 302: long sentence "We then compared the performance of cell estimates obtained applying our previously developed library with six cell-types available in FlowSorted. Blood.EPIC and optimized with the L-DMR IDOL library to the older 450 K array in artificial blood mixtures with predefined cell proportions. Four" Please clarify, suggest break down into shorter sentences.

Line 313: "HIV, HBV, and HBC." Please explain these acronyms. Do you mean HCV

Line 323: "The reconstruction mixtures were determined by randomly generating proportions from a twelve-component Dirichlet distribution." Is this an in silicon mixture or an experimental mixture?

Line 326: "Each mixture contained 1.2 µg of total DNA was estimated using the proportions included in Supplementary Table 2. The" Please clarify

Line 338: Wouldn't this sample selection based on predicted purity by your model potentially induce overfitting? Discuss

Line 352: "and SNPs derived from Infinium Type I probes were added using the total signal intensities." Why did you add SNPs? Please clarify

Line 354: "Probes corresponding to CpGs" or "measurements corresponding to probes"? The EPIC only has ~870k CpGs

Line 376: Can you show this elbow in a supplementary figure (not referenced)?

Reviewer #1 (Remarks to the Author: Reproducibility):

Work shows framework for reproducibility, given incorporation into Bioconductor pipelines presented. Sufficient reproducibility details and analyses are provided.

Reviewer #2 (Remarks to the Author: Overall significance):

Earlier, Salas et al. created a deconvolution library (a set of CpG markers) based on the Illumina 450K array DNA methylation data that can be used to discriminate six major cell types (CD4+, CD8+, NK, B, monocytes, and neutrophils) in blood. In the current work, the authors generated a new deconvolution library based on their Illumina 850K array data on 12 immune cell types to augment the cell type discrimination capability from 6 to 12. The 12 immune cell populations were isolated and purified from the PBMCs of 56 healthy donors. This new DNA methylation dataset is a useful resource not only for deconvolution of PBMC samples but also for other applications involving DNA methylation.

Reviewer #2 (Remarks to the Author: Strength of the claims):

My main concern about the work is how the validation of the libraries was carried out. One of the validation datasets was the cell mixtures of the 12 populations which is fine. However, it is not clear if any of the 56 samples were set aside as an independent dataset for validation. A real dataset with known cell proportions (e.g., from FACS) is essential for this goal. The authors should use a subset of the 56 samples as the test samples or consider generating additional independent PBMC samples with both DNA methylation and cell proportion data for validation.

The authors carried out a series of validation analyses using five datasets for most of which the cell proportions are only known for the six major types, not the 12 types. Without knowing the true proportions of the 12 cell types in each sample in those datasets, it is hard to imagine how one could assess the performance of the libraries on discriminating those populations. Besides the cell population mismatch problem, I do not find results for some of the validation data (e.g., the publicly available glioma, umbilical cord data, and PBMCs). Some description of those datasets and summary results should be presented and discussed.

Lastly, the authors also applied their libraries to 12 publicly available datasets with DNA methylation data from healthy controls and patients with diseases such as sclerosis or rheumatoid arthritis, and from pairs of monozygotic twins and dizygotic twins. Minimal descriptive results were presented for each dataset. It would be helpful to elaborate whether their analysis results recapitulated known biology or provided new insight on the underlying biology.

I also have issues with the Figures:

- Figure 1, legend is grossly simplistic. Some elaboration on each module would be needed.
- Figure 2A (top panel), it is not clear what the two proportional plots are referring to. The X-axis does not have label.
- Figure 2B, I would not plot both training and testing results in the same plot. What were the training and testing data used in this analysis? Were the training data the 56 samples and the testing data the cell mixtures? How were the cell mixtures generated? A supplementary table listing the proportions of each cell type might be good. Why were there only a few data points in each plot?
- Figure 3, some of the hypermethylated CpGs show little differential methylation patterns among the cell types. It is not clear how discriminative those CpGs are. What is the relevance of CpG genomic context, function context, and transcriptional regulatory elements of the CpGs in the context of cell type discrimination? I am not sure how informative those data are.
- Figure 4, why did the authors only plot data for the 6 major cell types and not the 12? The predicted and true proportions of the 12 immune cell types for the cell mixtures are available.
- Figure 5, how many of the CpGs in the libraries have also being identified by others as markers for deconvolution?

Minor comments

It is somewhat puzzling that the number of CpGs that overlap between the “legacy” (450 IDOL-ext) and EPIC IDOL-Ext libraries is rather low (300 out of 1500). The low overlap appears to be true for other comparisons (Table 1). Any explanation would be helpful.

Line 52, I would describe CIBERSORT as “support vector regression” rather than “support vector machine”.

How was the number of CpGs in each library (e.g., 1200 in EPIC IDO-Ext) determined, based on an arbitrary cutoff or some kind of feature selection?

I think it would be useful to include pDCs (plasmacytoid dendritic cells) in the panel should the authors consider to further expand their cell types in the future.

Reviewer #2 (Remarks to the Author: Reproducibility):

See my comments above. I think it can be improved for reproducibility.

Author rebuttal, first version:

Annotated Reviewer Reports

The editor has included some additional comments on the specific points raised by the reviewers below. However, please note that all points should be addressed in a revision, even if the editor has not specifically commented on them.

Reviewer #1		
Reviewer #1	This reviewer has not chosen to waive anonymity. The reviewer's identity can only be shared with representatives of an established journal editorial office.	
Reviewer #1 expertise Summarised by the editor	This reviewer has expertise in computational tool development, including techniques to analyze DNA methylation or deconvolute bulk datasets	
Editor's comment about this review	While this Reviewer provided positive feedback regarding the study design and potential resource value, they highlighted the need to further discuss potential advantages or limitations of this approach and offer several suggestions to improve analyses of tissue-specific methylation patterns. Based on feedback from this reviewer, we believe it would also be necessary (for consideration at Nature Communications) to discuss any potential limitations regarding the generalizability of this approach based on the demographic composition of the source cohort.	
Reviewer #1 comments		
Overview	The authors present a new method for cell type deconvolution of DNA methylation data, greatly expanding the number of cell types from previous algorithms, including Houseman and other methods. This paper represents a significant advance. The methodology is robust and the results are convincing. This paper and method promises broad application in the DNA methylation community, as virtually every human blood DNA methylation array study is likely to benefit from this software. Work and conclusions are original and solid. Previous work is appropriately credited, and appropriate context is provided. Paper is likely to be highly cited and influence the entire EWAS field.	
Specific comments		
#	Reviewer comment	Editorial comment/ Answers to reviewers and editors
Major Concerns		

1	Line 144: A lot of different consortium data includes both EPIC and 450k. Is there an approach here that can be applied on all the data. What about a common analysis/IDOL that includes both arrays? Is there a solution for this problem (could be valuable in some contexts)? Is this possible?	Editor comment: These points would only have to be speculated on in the Discussion for consideration at Communications Biology or Nature Communications. Answer: Thank you for pointing this possible scenario; we advise the researchers to evaluate their data quality before using any deconvolution approach to avoid potential technical biases (e.g., examining the library probe call rate and possible batch effects in the data). Although it is possible to use the IDOL-450k library for both, the accuracy is marginally better when using the IDOL-EPIC library probes. We have added a statement about that in the manuscript results: "We included some sensitivity analyses to compare the 450k IDOL-Ext vs. the EPIC IDOL-Ext libraries when using EPIC or a combination of EPIC and 450k legacy arrays. As shown in Supplementary Fig. 4, both libraries provide an accurate estimation for the 12 cell types. There are minor variations in Bmem and Bnv estimations, with more precise estimates using the EPIC IDOL-Ext (absolute mean difference 0.08%-EPIC vs. 0.48%-450k, and 0.23-EPIC% vs. 0.52%-450k, respectively), and CD4mem and Treg with more precise estimations using the 450k IDOL-Ext (absolute mean difference 0.62%-EPIC vs. 0.07%-450k, and 0.6-EPIC% vs. 0.34%-450k, respectively), see Supplementary Fig. 17 for additional information." And in the Discussion: "Both EPIC IDOL-Ext and 450k IDOL-Ext are accurate, with the former requiring fewer probes for the estimation. In complex problems, such as EWAS using EPIC and 450k technologies simultaneously, we recommend using the more straightforward approach for the data using the 450k IDOL-Ext given the minimal variation observed (<1%) in the estimates. We advise the researchers to evaluate their data quality before using any deconvolution approach to avoid potential technical biases (e.g., examining the library probe call rate and possible batch effects in the data)."
2	Line 240: "Our libraries have high accuracy and minimal bias when compared to flow cytometry gold standard data." Please reference some numbers/estimates here. Progress is shown and this method represents a significant advance but no method is perfect. Please discuss pros and cons.	Editor comment: As mentioned in Point #4 below, proper discussion of the context, advantages, and limitations of the method would be necessary for further consideration at Communications Biology or Nature Communications. Answer: Thank you for pointing this out. We have added a few lines in the Discussion to explain the accuracy and limitations: "Our libraries have high accuracy and minimal

		bias compared to flow cytometry, considered the “gold standard” data (Figure 4). For most cell types, that accuracy represents a difference of less than 3% in the estimations. However, more considerable differences (>4%) are observed in some datasets for the most abundant cell-types (neutrophils and total CD4T, Supplementary Figs. 6 and 7) and for CD8Tmem (Supplementary Fig. 6). This error results from limited purity in this cell component and possibly differences in gating processes during the count splitting. For the CD8mem compartment, it is also possible that unaccounted residual heterogeneity is present, TEMRA, effector memory, central memory, NKT cells, and other less characterized CD8Tmem subclasses. The new libraries will be important where specific memory and naïve cell compartments are interrogated for disease, pathogenesis, or exposure.”
3	Line 248: Main comment: Results for specific genes are interesting but given that many of these DNA methylation sites are located in open sea regions and enhancers, a chromatin based analysis approach that does not depend on genes would be appropriate. Why not use eFORGE/eFORGE-TF or another motif or chromatin analysis method to validate independently that these are tissue-specific elements (ideally using an enhancer mark e.g. H3K4me1)? An example of such an analysis is appended as a pdf. The MSigDB analysis is interesting but paper would greatly benefit from an agnostic, chromatin-based method given the amount of enhancer-located sites.	Editor comment: Examination of tissue-specific elements using eFORGE or a related tool would be necessary for consideration at Communications Biology or Nature Communications. Answer: Thank you for this recommendation; we have approached the problem based on cell-specific probes. When cell-specific marks are available, the eForge consistently enriches regulatory regions for the specific cell types. We added the following sentences to the results and a related figure to the supplementary materials: “We also used eForge (https://eforge.altiusinstitute.org/) to identify whether the components of the library were related to particular primed enhancer histone marks (H3K4me1). When analyzing probes specific for the 12 cell types, all subsets were enriched for blood components, but the top significant results were consistent with the specific cell-type: E029 Monocytes with monocytes (Supplementary Fig. 8 panel f), E032 B cells with Bnv, and Bmem (panels b and a), E033 T cells with CD4nv, CD4mem, Treg, CD8nv and CD8mem (panels i, h, j, l, and k, respectively) and E046 Natural killers with NK (panel g). There were no specific datasets to compare the granulocytes (Bas, Eos, Neu), where the signal was less straightforward (panels c, d, and e).”
4	Line 262: Main comment: In what way does the IDOL method compare with other methods, for example robust partial correlations? What are the advantages of this iterative method? Discuss	Answer: The IDOL method is designed for iterative optimization of CpG probe selection for the deconvolution library itself and not for determining cell-type proportions. However, we have added some results comparing the constrained projection/quadratic programming (CP/QP) deconvolution method we apply with RPC and CIBERSORT. “Results using other statistical procedures for cell

		deconvolution, CIBERSORT, and robust partial correlations, were compared and are illustrated in Supplementary Fig. 18 . Results were generally similar using the different methods, though slightly lower accuracy was observed for CIBERSORT compared to CP/QP or robust partial correlations.”
Minor Concerns		
5	Line 26: Retrospective, do you mean "previous"?	Editor comment: Most of these minor comments reflect grammatical errors or unclear points of discussion. Please carefully proofread for clarity and grammatical errors. Answer: This has been corrected.
6	Line 31: "Immune system" or "epigenetics of the immunesystem"?	Answer: This has been corrected. We meant immune profiles (derived from the library) in general. We hope this is clearer now.
7	Throughout the manuscript: Continued use of the word "library" could potentially be confusing. As the authors will know, library is commonly used in experimental genomics with a slightly different angle and of course this is purely computational so context is different. There is a precedent in previous pipelines but potentially worth clarifying at the beginning of the text.	Answer: Thank you for the comment. We have added a clarification in the introduction: "Constrained projection/quadratic programming (CP/QP) employs purified cell types as reference samples to generate a "reference library," a matrix of differentially methylated sites among cell types, and yields highly accurate estimates of the underlying cell composition in mixed cell populations (e.g., peripheral blood) ⁹ . Previously established statistical deconvolution frameworks such as CP/QP, support vector regression (CIBERSORT), and robust partial regression (EpiDISH) have similar accuracy and precision in deconvolution estimates ¹⁰ ."
8	Line 80: Explain how you go from 12 to 19.	Answer: Thank you for pointing this out. We have added some text for context: "Our comprehensive library provides information across 12 different cell subtypes; depending on the hypothesis, categories could be collapsed at in seven additional higher branches (T-cells, B-cells, CD8T, CD4T, granulocytes, lymphoid and myeloid), Figure 1 panel c , resulting in 19 relative cell-type proportions and 19 cell-counts (derived data using the complete cell blood counts or flow cytometry)"
9	Line 106: Why is this lower number in the range (85.7) not included/mentioned in the results discussed immediately afterwards (we go from mean to median, maybe could be clarified and focus on just one measure)?	Answer: We understand the reviewer's concern about using different descriptive statistics. However, here, we are trying to emphasize that the CD8mem cell purification is problematic and could be a source of bias in the library. We would also prefer to keep the mean ranges and median IQR range for the reader, as the distributions are not shown in the main text but only as supplementary figure 1. These will be useful for the reader to understand the inner and outer fences without the figure. We have also modified the sentence to clarify that CD8mem represents the lowest range in the distribution 85.7%: "and CD8mem cells had the lowest DNA methylation purity (range: 85.7-98.5%, median DNA methylation purity: 93%, IQR: 88.3-97.8%)."

10	Line 117: "slide beadchip" as a remainder of technical variation. I assume this is later removed. Please discuss/mention this.	Editorial comment: We believe the Reviewer is asking you to clarify whether (and how) technical variability was accounted for during analysis. Answer: Unfortunately, the effect of Slide was not possible to eliminate in this experiment. With twelve cell types and only eight samples per slide there is a limitation of the ability to completely randomize cell type with respect to slide. To build the extended deconvolution libraries we analyzed data samples collected across 26 slides. Cell-type explains more than 90% of the variance in the principal component regression for the first principal component of the beta-values in all the cell samples, and it is >80% of the variance for the top 4 principal components regressions. Slide, in contrast, explains less than 50% of the variance in a principal regression model as the unique explanatory variable and, when added to cell-type, only increases 1% of the variance explained in the mutually adjusted model. Although the potential for technical variability by slide may be minimally present, most of the variance is explained by cell-type. We have added the following lines acknowledging this limitation: "Due to the distribution of the 12 cell types across 26 different slide chips, some residual experimental variance cannot be corrected in the modeling. As such, we strived for the highest quality data to eliminate most of the additional technical variability in the experiment."
11	Line 133: I am familiar with CpG Island shores but still not sure if north or south component has much relevance here, please explain and if not clear suggest focusing just on describing "CpG Island shores" as north and south might be confusing to non-experts.	Editorial Comment: Please either clearly define the concept of "north" and "south" shore regions, or limit the discussion to the broad concept of CpG island shores. Answer: We agree with the reviewer, we have simplified the discussion of North (5') and South (3') shores to the broader termed Shores and Shelves.
12	Line 150: Regarding the genomic context, it is not clear if the probe distribution by genomic context for IDOL libraries is any different from the array from which the data was selected... comparison to background probes on the array and some statistical test could potentially be enlightening in this respect.	Answer: Thank you for this comment. We have added results and new related Supplementary Table 5. Briefly, when testing the enrichment of the different genomic context features versus the background probes used for the selection of the probes, we observed significant enrichment of CpGs in open sea and enhancer regions and a depletion CpGs in CpG islands. We have also added this information to the text.
13	line 168: Was the flow cytometry done by this study or by a previously published study? Please clarify.	Editorial comment: Please provide a citation (not just the accession), if relevant. Answer: Thank you for this commentary; Nature Methods formatting guidelines restricted inclusion of all references, and we have updated this version to include the references to FCM that were part of our previous studies published in 2016 and 2018. We have also clarified the content and setting of these experiments "In five samples (Figure 4a, GSE110530) from a healthy male subject in his forties, who was followed longitudinally

		for 400 days, we observed a strong correlation between the deconvolution estimates and FCM measurements, with a maximum root mean square error- RMSE of 2.15% for CD4T cells.¹⁹ “The 450k IDOL-Ext library was also validated using the GSE77797 dataset with six anonymous adult volunteers [5 males, epigenetic age mean (sd): 34.1(11.4) years] with FCM information (Figure 4c) and six artificial mixtures with six known cell-types (Neu, CD8T, CD4T, Bcell, Mono, NK). The information from this dataset was used in the original IDOL manuscript (Figure 4d).¹²
14	Line 187: Why was the component of nucleated red bloodcells added to the myeloid granulocyte component specifically?	Answer: The reason was mostly biologically driven, as hierarchically erythromyeloid lineages have a shared common progenitor. That can be reconstructed using DNA methylation as tested previously by Farlik et al. We also observed that the nucleated red blood cells clustered with the granulocytes and, in particular, the basophils using the library markers. We have added some sentences and a supplementary figure to support it. “The rationale for adding the erythroblasts to the granulocytes is based on the known biology that establishes the closest lineage to the erythroid-myeloid precursor.²⁶ This is reinforced by clustering of nucleated red blood cells with basophils and eosinophils using the markers in EPIC IDOL-Ext (Supplementary Fig. 7a).”
15	Line 216: A lot of proportional findings for different diseases are reported; can one be explained in depth? Suggest to focus on at least one and describe biological interpretation in depth.	Answer: Although we feel this goes beyond the scope of the manuscript, we now include additional discussion of the Rheumatoid Arthritis results. “We applied this library to several public datasets, comparing cases and controls for several pathologies. Our results are concordant with known pathophysiology and immune responses for these diseases: increased proportions of abnormal naive B cell populations in the blood in multiple sclerosis patients,⁴⁷ pancytopenias with a predominance of neutrophils in radiotherapy plus chemotherapy in breast cancer,⁴⁸ predominancies of neutrophilia and lymphopenia in COVID-19 patients that did not show remission⁴⁹. However, the limited sample sizes of these data preclude more generalizable conclusions for most of the findings. We will only discuss the most extensive dataset available of rheumatoid arthritis patients vs. controls. This dataset, published by Liu et al.,⁵⁰ has explored how some immune cell components (monocytes) are associated with specific DNA methylation changes in the cases who were not receiving any treatment at the time of blood collection. We observed a reduction in monocyte proportions in rheumatoid arthritis patients compared to controls (P-value=1.1 E-5). Paradoxically in our analysis, we found a non-statistically

		significant increase in Tregs in cases compared to controls (P-value=0.06), which are related to disease activity or response to medications in other studies. ⁵¹ These differences are essential to understand the activity of the disease and potentially for classification of the heterogeneous subjects when compared to healthy controls.”
16	Figure 3: As the authors know, DHSs are open chromatin, so is "open chromatin" used for ATAC-seq data? Discuss	Answer: We completely agree with the reviewer, and this deserves a more extended discussion beyond the scope of this manuscript as these open chromatin features are also cell-type specific. However, here we are not using cell-specific open-chromatin information. Instead, we are using the ENCODE summary data provided in the latest EPIC annotation file. For simplicity for the readers, we are only using the information contained in the annotation file for this figure. We have clarified this in the caption. “Information about CpG island gene context, Phantom5 enhancer information, DNase Hypersensitivity sites (DHS), Open chromatin, and annotated transcription factor binding sites (TFBS) from ENCODE included in the Illumina annotation file are summarized using the row ribbons.”
17	Figure 4A: Why R ² 0.16 for monocytes?	Editorial comment: Please clarify why there is such a low correlation for monocytes in relation to the other cell types in this panel. Answer: This is the result of a low variability for the cell-type. For those cell types with lower distribution width, any minor deviation from the line will impact the coefficient of determination. We had corrected some minor problems with our normalization, so this number has been updated to 0.19. We have added the following: “Of note, the coefficient of determination was lower for monocytes (0.19) and NK (0.66), Figures 4a and 4c , respectively, using the EPIC IDOL-Ext library. This decrease is driven by the narrower range of distribution of these cells in the FCM samples. Consequently, any minor deviation in the estimation is magnified even with the high precision reflected by the RMSE.”
18	Figure 4C: Discuss R ² 0.65 for NK cells.	Editorial comment: Please clarify why there is such a low correlation for NK cells in relation to the other cell types in this panel. Answer: Please see our previous answer. Also, please notice that this number has also been updated to 0.66.
19	Line 227: change in immune proportions with age. Can part of the literature on this be referenced?	Answer: We have included some references (Simon, 2015, Zheng, 2020) for the immune changes and literature on DNA methylation changes with aging. “Extensive literature has explored the changes in immune proportions with aging, ^{27,28} but it has not been

		systematically explored using DNA methylation. ²⁹⁻³¹
20	Line 239: ">100,000 samples" Where is this number from?	Answer: We have clarified that these are all the available samples in GEO as July 2021 for both 450k and EPIC platforms. "The Gene Expression Omnibus repository, as of July 2021, includes publicly available data from ~148,325 samples for both platforms. ³² " We also added the Maden et al. reference.
21	Line 256: Genes or probes?	Answer: We apologize for this typo. We have corrected the sentence to probes.
22	Lines 264/265: I'm not sure what EPX absence proves maybe this should be explained further or contextualized?	Answer: We have included a clarifying sentence. "For example, four (EPIC) and five (450k) EPX probes were selected in the extended libraries, in contrast to none by the minfi pickCompProbes procedure. EPX is the gene containing the information for the eosinophil peroxidase protein, which is essential for eosinophil function and solely expressed in this blood cell type. ^{40,44} "
23	Line 268: "than" or "from"?	Answer: We have rewritten the sentence as: "Particularly intriguing for umbilical cord blood, nucleated red blood cells interfered with more precise estimations than available FCM information."
24	Line 273: Seems like a comma is missing.	Answer: we have corrected this sentence as "These cells are involved in multiple pathologies, including allergies and asthma, and through these libraries, they can now be estimated."
25	Line 274: can or "they can"?	Answer: Please see the previous answer
26	Line 276: "Thus careful interpretation based on those estimates alone is prudent." Please clarify.	Answer: We have rearranged the sentence at the end of the paragraph and added the following: "Thus careful interpretation based on those estimates alone is prudent as they may signal residual cell signals of other cell-types not included in the library."
27	Line 282: "a careful assessment of any high basophil signalis warranted" Could the tool include a flag/warning for these high basophil signals as default?	Answer: We can incorporate the flag. However, this could be problematic for some users. Although it is more common to use whole blood, it is also expected that many end-users will aim to estimate subfractions of cells from PBMCs, granulocytes, or other isolated fractions.
28	Line 289: "controlling" or "controlling for"?	Answer: This has been corrected.
29	Line 289: "effects", do you mean "DNA methylationchanges"?	Answer: This has been corrected.
30	Line 290: "for" or "for in"?	Answer: This has been corrected.

31	Line 292: "to" or "for"?	Answer: This has been corrected.
32	Line 302: Long sentence "We then compared the performance of cell estimates obtained applying our previously developed library with six cell-types available in FlowSorted. Blood.EPIC and optimized with the L-DMR IDOL library to the older 450 K array in artificial blood mixtures with predefined cell proportions." Please clarify, I suggest breaking this statement down into shorter sentences.	Answer: Thank you for pointing this error. We have split the run over sentence into shorter fragments: "We applied the IDOL method to identify optimal Leukocyte -Differentially Methylated Regions (L-DMR) libraries using a testing set of six artificial mixtures containing the 12 cell-types. Artificial mixtures (also referred to as reconstructions) consist of DNA from purified isolated cell types, representing mock blood samples of known, predefined cell proportions. Six additional testing artificial mixtures were employed to corroborate the performance, and 12 independent artificial mixtures derived from a set of 12 isolated cell-types not utilized in the training or testing, were used to validate the results. We compared the performance of cell estimates obtained applying our previously developed six cell-type (available in Bioconductor as "FlowSorted. Blood.EPIC") and optimized an additional L-DMR IDOL library limiting the probes to those available in the older 450 K array, and again compared the performance using the training, testing, and validation datasets."
33	Line 313: "HIV, HBV, and HBC." Please explain these acronyms. Do you mean HCV?	Answer: This has been corrected. "They were negative for Human Immunodeficiency Virus-HIV, Hepatitis B Virus-HBV, and Hepatitis C Virus-HCV"
34	Line 323: "The reconstruction mixtures were determined by randomly generating proportions from a twelve- component Dirichlet distribution." Is this an in silico mixture or an experimental mixture?	Answer: These are experimental mixtures, as mentioned in the next sentence. However, the proportions were generated using a Dirichlet distribution. "Twenty-four artificial mixtures were determined by randomly generating proportions from a twelve-component Dirichlet distribution. Each mixture of 1.2 µg total DNA was generated from isolated cell DNA using the proportions in Supplementary Table 3."
35	Line 326: "Each mixture contained 1.2 µg of total DNA was estimated using the proportions included in Supplementary Table 2." Please clarify.	Answer: This corresponds to the total DNA mass used per sample.
36	Line 338: Wouldn't this sample selection based on predicted purity by your model potentially induce overfitting? Discuss.	Answer: Not at all. This is a quality control measurement using a statistical proxy. The main problem when using sorted cells is the presence of unintended contaminants. Cell surface markers that are shared across cell-types may generate imperfect cell isolation. In our experience, disagreements in the DNA methylation purity are related to problems in cell isolation. This measure is not used in any of the models downstream.
37	Line 352: "and SNPs derived from Infinium Type I probes were added using the total signal intensities."	Answer: This is a technique for some probes with SNP interference (Zhou, 2017), that could generate spurious channel switching, but also can be used to derive genetic ancestry markers. We have clarified that in the text: "The

	Why did you add SNPs? Please clarify,.	package contains an RGChannelSet R object processed using SeSaMe in which probes showing channel switching were corrected and SNPs derived from Infinium Type I probes were added, using the total signal intensities, to the technical control SNPs for genetic ancestry evaluation." The expanded SNPs (plus the 59 technical SNPs) are included in Supplementary Fig. 2.
38	Line 354: "Probes corresponding to CpGs" or "measurements corresponding to probes"? The EPIC only has ~870k CpGs	Answer: This includes Infinium I (each with two probes (one per channel in two different addresses) and Infinium II probes (one probe for both red and green channels in one address). Those probes interrogate 866,091 sites (CpGs and CpGs). For additional information, please see the Illumina technical note https://www.illumina.com/documents/products/technote/technote_hm450_data_analysis_optimization.pdf
39	Line 376: Can you show this elbow in a supplementary figure (not referenced)?	Answer: We have included Supplementary Table 3 and included the elbow in bold.
40	Reproducibility: Work shows framework for reproducibility, given incorporation into Bioconductor pipelines presented. Sufficient reproducibility details and analyses are provided.	Answer: Thank you.

Reviewer #2

Reviewer #2	This reviewer has not chosen to waive anonymity. The reviewer's identity can only be shared with representatives of an established journal editorial office.	
Reviewer #2 expertise Summarised by the editor	This reviewer has expertise in computational tool development and deconvolution of bulk sequencing data.	
Editor's comment about this review	This Reviewer also offers positive feedback regarding the resource value, but raises an important concern regarding whether the library was appropriately validated in an independent dataset. They also raise concerns about data availability (namely, presenting summary results) that should be addressed in a revision.	
Reviewer #2 comments		
Overview	Earlier, Salas et al. created a deconvolution library (a set of CpG markers) based on the Illumina 450K array DNA methylation data that can be used to discriminate six major cell types (CD4+, CD8+, NK, B, monocytes, and neutrophils) in blood. In the current work, the authors generated a new deconvolution library based on their Illumina 850K array data on 12 immune cell types to augment the cell type discrimination capability from 6 to 12. The 12 immune cell populations were isolated and purified from the PBMCs of 56 healthy donors. This new DNA methylation dataset is a useful resource not only for deconvolution of PBMC samples but also for other applications involving DNA methylation.	
Specific comments		
#	Reviewer comment	Editorial comment
Major Concerns		
1	My main concern about the work is how the validation of the libraries was carried out. One of the validation datasets was the cell mixtures of the 12 populations which is fine. However, it is not clear if any of the 56 samples were set aside as an independent dataset for validation. A real dataset with known cell proportions (e.g., from FACS) is essential for this goal. The authors should use a subset of the 56 samples as the test samples or consider generating additional independent PBMC samples with both DNA methylation and cell proportion data for validation.	Editorial comment: Inclusion of an independent dataset (including FACS data) would be necessary for consideration at Communications Biology or Nature Communications. Answer: Thank you for your suggestion. We ran a new independent validation experiment with cell mixtures for all 12 cell types using DNA from cells not included in our library training or testing. We have reorganized Figure 2 to show the testing (panel a), training (panel b), and independent validation sets (panel c). The results are highly concordant with a high coefficient of determination. We have clarified this in the methods: "We applied the IDOL method to identify optimal Leukocyte -

		Differentially Methylated Regions (L-DMR) libraries using a testing set of six artificial mixtures containing the 12 cell-types. Artificial mixtures (also referred to as reconstructions) consist of DNA from purified isolated cell types, representing mock blood samples of known, predefined cell proportions. Six additional testing artificial mixtures were employed to corroborate the performance, and 12 independent artificial mixtures derived from a set of 12 isolated cell-types not utilized in the training or testing, were used to validate the results..” And in the results section: “The library was further validated using a set of 12 artificial mixtures obtained from independently isolated samples (purity>90%) not used in the library construction (Figure 2c), with a high coefficient of determination and low RMSE. We observed some slight increase in the error estimations for CD4mem (RMSE= 3.64) with compensatory decreases in the Treg, CD4nv, and CD8mem compartments.”
2	The authors carried out a series of validation analyses using five datasets for most of which the cell proportions are only known for the six major types, not the 12 types. Without knowing the true proportions of the 12 cell types in each sample in those datasets, it is hard to image how one could assess the performance of the libraries on discriminating those populations. Besides the cell population mismatch problem, I do not find results for some of the validation data (e.g., the publicly available glioma, umbilical cord data, and PBMCs). Some description of those datasets and summary results should be presented and discussed.	Editorial comment: Please include this data (at a minimum, the summary results) along with the revised manuscript. The Reviewer also mentions a broader point about how datasets are introduced in the Results, and quickly passed over. Please ensure that you thoroughly discuss the relevance and any insight provided through the analysis of each dataset. Answer: Thank you for your suggestions. We have run a new independent experiment including 12 independent validation artificial mixtures using DNA from cells not included in our library as mentioned in our previous point. These samples include the 12 cell-types in different proportions. We included the FCM data from different sources as potential external validation, and we agree with the reviewer about all the limitations of the heterogeneous experiments. These are complex datasets used for other research purposes beyond this manuscript. We also want to emphasize that flow cytometry information has some limitations as a validation set, as the gating can be selected differently by different analysts or automatic algorithms providing different answers to the grouping. Also, in real world mixtures the presence of additional cell types can be observed that can generate additional noise and higher error rates for the algorithm. We have also added the demographic information,

and the experiments per each sample in our results:

We transcribe some of the new text below:

“The EPIC IDOL-Ext library was validated using samples with blood cell counts from flow cytometry (FCM) and by using an independent set of artificial mixtures from the Gene Expression Omnibus (GEO) (**Figure 4**). In five samples (**Figure 4a**, GSE110530) from a healthy male subject in his forties, who was followed longitudinally for 400 days, we observed a strong correlation between the deconvolution estimates and FCM measurements, with a maximum root mean square error- RMSE of 2.15% for CD4T cells.¹⁹”

“Using independent artificial mixtures from the previously published FlowSorted.Blood.EPIC library with six known cell-types (Neu, CD8T, CD4T, Bcell, Mono, NK; **Figure 4b**, GSE110554), ...”

“The 450k IDOL-Ext library was also validated using the GSE77797 dataset with six anonymous adult volunteers [5 males, epigenetic age mean (sd): 34.1(11.4) years] with FCM information (**Figure 4c**) and six artificial mixtures with six known cell-types (Neu, CD8T, CD4T, Bcell, Mono, NK). The information from this dataset was used in the original IDOL manuscript (**Figure 4d**).¹²”

“As a proof-of-concept, we explored how the cell proportion estimates from the EPIC IDOL-Ext library could recapitulate cell counts using FCM cell count information from the GSE112618 dataset (**Supplementary Fig. 5**).¹⁹ This dataset included six subjects [five males: mean (sd) age 42.8 (10.6) years] with counts for nine cell types (Bas, Bcell, CD4T, CD8T, Eos, Neu, Mono, NK, and Treg). Additional validation sets were analyzed, including glioma patient blood and the cytometric information for the Reinius dataset¹⁷. A group of 76 glioma patients (GSE180683) was used to evaluate the information from T-cell subsets. The patients were 51% male with a mean (sd) epigenetic age (Horvath) of 54.7 (14.1) years.”

“Finally, we explored whether these libraries could be applied for the deconvolution of umbilical cord blood samples. We used the Jones et al. GSE127824 dataset (450k) with known FCM information for seven cell-types (erythroblasts-nucleated red blood cells, granulocytes, CD8T, CD4T, Mono, NK, and Bcell).²⁵”

“In contrast, in an experiment of 12 umbilical cord

		blood artificial mixtures (GSE180970) from healthy anonymous newborns acquired from commercial vendors with only six cell types (granulocytes, CD8T, CD4T, Mono, NK, and Bcell),...”
3	Lastly, the authors also applied their libraries to 12 publicly available datasets with DNA methylation data from healthy controls and patients with diseases such as sclerosis or rheumatoid arthritis, and from pairs of monozygotic twins and dizygotic twins. Minimal descriptive results were presented for each dataset. It would be helpful to elaborate whether their analysis results recapitulated known biology or provided new insight on the underlying biology.	Editorial comment: Please expand the Results and Discussion to comment on new or confirmatory results from each of these datasets. Answer: We have expanded the results for RA, and explained that the limited sample sizes limit our ability to generate conclusions for other datasets. “We applied this library to several public datasets, comparing cases and controls for several pathologies. Our results are concordant with known pathophysiology and immune responses for these diseases: increased proportions of abnormal naive B cell populations in the blood in multiple sclerosis patients,⁴⁷ pancytopenias with a predominance of neutrophils in radiotherapy plus chemotherapy in breast cancer,⁴⁸ predominancies of neutrophilia and lymphopenia in COVID-19 patients that did not show remission⁴⁹. However, the limited sample sizes of these data preclude more generalizable conclusions for most of the findings. We will only discuss the most extensive dataset available of rheumatoid arthritis patients vs. controls. This dataset, published by Liu et al.,⁵⁰ has explored how some immune cell components (monocytes) are associated with specific DNA methylation changes in the cases who were not receiving any treatment at the time of blood collection. We observed a reduction in monocyte proportions in rheumatoid arthritis patients compared to controls (P-value=1.1 E-5). Paradoxically in our analysis, we found a non-statistically significant increase in Tregs in cases compared to controls (P-value=0.06), which are related to disease activity or response to medications in other studies.⁵¹ These differences are essential to understand the activity of the disease and potentially for classification of the heterogeneous subjects when compared to healthy controls. It is difficult to predict the extent of discoveries made possible with the extended capability for immune profiling we provide. Our method gives higher resolution than clinical approaches and allows for more scalable, efficient, standardized immune profiling compared with flow cytometry, and has the added benefit of applying to archival specimens. Although we have only begun to demonstrate the potential of our approach, we do not doubt that the research community will leverage this

		method to provide new insights in the relation of immune profiles in human health and disease in exciting and meaningful ways that impact public health and translational medicine.”
Comments Regarding Figures		
4	Figure 1: The legend is grossly simplistic. Some elaboration on each module would be needed.	Editorial comment: Please provide a more descriptive title. We would also recommend breaking the figure into multiple subpanels (say, Fig. 1a-c) that could be further described in the legend. Answer: Thank you for your suggestion; we have changed Figure 1 entirely and added a complete description for the three panels “Figure 1. Process summary. Panel a: 12 cell-types were acquired commercially, their DNA was isolated, and DNA methylation was measured using the Illumina HumanMethylationEPIC (EPIC) microarray. Two libraries were identified using an iterative process named IDOL (IDentifying Optimal Libraries, Koestler et al. 2016) using artificial mixtures as ground truth. The two extended (ext) libraries were designed for microarray data derived from the EPIC array (EPIC IDOL-ext) or legacy data derived from the previous Illumina HumanMethylation450k array (450k IDOL-ext). Panel b: samples with variable amounts of leukocytes are arrayed using any of the two validated microarray technologies. Using the appropriate library for the microarray, a cell mixture deconvolution is performed using the constrained projection/quadratic programming (CP/QP, Houseman et al. 2012). *Optionally, leukocyte counts can be collected for downstream analyses. (c) The primary results of the deconvolution are the twelve cell types of the library. These results could be aggregated at different levels for different hypotheses. Two sets of derived results are possible: 1) if total leukocyte counts are available cell-type-specific counts may be inferred, or 2) cell ratios and proportions are used to evaluate immune cell shifting between the different cell-type subpopulations (only a few examples illustrated here)”
5	Figure 2A (top panel): It is not clear what the two proportional plots are referring to. The X-axis does not have a label.	Answer: Thank you for pointing this out. We have added the labels for the piled barplots. Each bar represents a sample (x axes), and the y axes represent the Cell % per sample. The stacked barplots add to 100% as the information is known in each sample.

6	Figure 2B: I would not plot both training and testing results in the same plot. What were the training and testing data used in this analysis? Were the training data the 56 samples and the testing data the cell mixtures? How were the cell mixtures generated? A supplementary table listing the proportions of each cell type might be good. Why were there only a few data points in each plot?	Answer: Thank you for the suggestions. We have separated this figure and added the independent validation. Supplementary Table 2 contains the proportions for each sample. Please notice that for some cell-types in the training dataset the proportions were zero (basophils and Tregs), this was done purposely as these cell-types can be zero in actual samples. The dots in those cases overlap in the graph. We have also clarified in the methods how the artificial mixtures were generated: “Twenty-four artificial mixtures were determined by randomly generating proportions from a twelve-component Dirichlet distribution. Each mixture of 1.2 µg total DNA was generated from isolated cell DNA using the proportions in Supplementary Table 3. The isolated cell DNA and those of the artificial mixtures were bisulfite converted and processed according to the Illumina protocols...”
7	Figure 3: Some of the hypermethylated CpGs show little differential methylation patterns among the cell types. It is not clear how discriminative those CpGs are. What is the relevance of CpG genomic context, function context, and transcriptional regulatory elements of the CpGs in the context of cell type discrimination? I am not sure how informative those data are.	Answer: thank you for pointing this out. The algorithm selects probes that are useful in the multivariable deconvolution model despite not all cell-to-cell comparisons of differential methylation having the same distribution of differential methylation. Optimal libraries for six or more cell types are composed of hundreds to more than a thousand CpGs, and we do investigate the discriminative capacity of single CpGs. Nonetheless, many hypermethylated CpGs with lower differential methylation appear most specific to discriminating among Treg, CD4mem and CD8mem albeit to a lesser extent than other sites. The genomic/functional context for the library sites is shown in Figure 3. We also include tests of enrichment indicating enrichment of enhancer and open sea CpGs and a depletion of CpG island CpGs. Future work on the relation of methylation with transcription includes transcriptional regulatory elements and is needed as the biology behind the probes represents not only the changes in methylation, but also other changes in transcriptional activity (enhancers, open chromatin, etc.) is likely to be informative.
8	Figure 4: Why did the authors only plot data for the 6 major cell types and not the 12? The predicted and true proportions of the 12 immune cell types for the cell mixtures are available.	Editorial comment: Please include data for all 12 celltypes. Answer: We apologize if this was unclear. We believe the plot may not have been clear because subtypes of cells are captured in pie plots which do not always have large enough fractions of pie to see due to the size of the pie plot. We have increased the size of the pie dots, to better show the proportions. In this case, we compare

		the collapsed categories with known information vs. the estimates derived from the extended library.
9	Figure 5: How many of the CpGs in the libraries have also been identified by others as markers for deconvolution?	Answer: We have compared only our results vs. the minfi algorithm, and our previous library published in 2018. The overlaps are detailed in table 1. We have not performed a comparison in the manuscript with other algorithms such as EpiDISH as the library is limited to 450k microarrays (only 315 out of 333 probes are present in the EPIC microarray), and the comparison will not be fair. However, from their EPIDISH DHS list only 14/333 of the EPIC IDOL-Ext and 24/333 in the 450k IDOL-Ext overlapped with our libraries.
Minor Concerns:		
10	It is somewhat puzzling that the number of CpGs that overlap between the “legacy” (450 IDOL-ext) and EPIC IDOL-Ext libraries is rather low (300 out of 1500). The low overlap appears to be true for other comparisons (Table 1). Any explanation would be helpful.	Answer: We understand the concern. We have to notice that the universes for selection were different. However, among the 459 sites from the EPIC IDOL-Ext, 72% overlapped with those selected in the 450k IDOL-Ext. So a majority of these 450k markers are shared between the libraries.
11	Line 52, I would describe CIBERSORT as “support vector regression” rather than “support vector machine”.	Answer: Thank you for pointing this typo. This has been corrected.
12	How was the number of CpGs in each library (e.g., 1200 in EPIC IDO-Ext) determined, based on an arbitrary cutoff or some kind of feature selection?	Answer: For the IDOL library selection, we performed an iterative process, looking for the elbow with better performance. The lowest RMSE and highest coefficient of determination were used as cut-off. The elbow is represented in Supplementary Table 3 , showing the change in the RMSE when reaching 1200 and 1500 probes, respectively, before degrading the signal.
13	I think it would be useful to include pDCs (plasmacytoid dendritic cells) in the panel should the authors consider to further expand their cell types in the future.	Editor comment: While we agree that expanding the panel would be useful, this point would not be necessary for Communications Biology or Nature Communications . Answer: Thank you for the suggestion. We agree this will be of interest. We continue to explore other potential cell-types. One limitation of additional cell types like pDCs is the isolation of sufficient cells/DNA to generate a relevant panel which requires a larger blood donor volume. We are currently evaluating experimental alternatives for those cases. We have added this sentiment in the conclusions and future directions “Future work includes validating the use of this library in methylation data from children and umbilical cord blood and expanding the libraries to additional cell subsets critical for some pathologies (e.g., dendritic cells).”

14

Reproducibility: See my comments above. I think it can be improved for reproducibility.

Editorial comment: We believe the Reviewer's concerns from reproducibility largely stem from the absence of summary statistics. Please also be sure to clearly state the source of each dataset used for analysis.

Answer: Thank you for your suggestions, we agree with the reviewer and apologize for our oversight of these important details. We have now summarized the demographic information of all the main datasets and the external applications sets in the results, and the granular information for the purified cell-types and the external applications (**Supplementary tables 2 and 7**). The details of each dataset are now included in the text (see answer to comment #2). All the datasets are or will be available upon publication in GEO or ArrayExpress to ensure reproducibility. We have also added the GEO tokens for the reviewers.

"Data availability: The main source dataset (12 cell types, testing, and training artificial mixtures) is hosted on GSE167998 and can be accessed using the token etqpgkouddybxyz. The Superseries GSE181034 can be accessed using the token azgnkemgvdylgz. This is composed of the following series GSE180683 (glioma samples, validations for T cell memory subsets), GSE180970 (umbilical cord artificial mixtures), GSE182379 (independent validation 12 cell-types artificial mixtures). All the datasets will be public upon the manuscript acceptance. Additional datasets analyzed in this manuscript are publicly available in GEO and ArrayExpress with accession numbers: GSE110554, GSE77797, GSE110530, GSE35069, GSE68456, GSE88824, GSE42861, GSE140038, GSE161778, GSE105018, E-MTAB-7069, GSE85042, GSE103189, GSE104778, GSE62219, GSE87571, E-MTAB-7309, GSE87571, GSE12163. The code used for this manuscript has been deposited in Zenodo doi: 10.5281/zenodo.5338513⁵⁹."

Reviewer comments, second version:

Reviewer #1 (Remarks to the Author: Overall significance):

Amendments are satisfactory and the manuscript has been significantly improved. A few minor comments:

- Re "Gating processes" can you explain/provide a reference?
- Explain TEMRA & other acronyms (important to explain every acronym)
- Provide references in TEMRA/NKT section
- Re point 27 consider to at least include an explanation on GitHub, or flag when the signal is above a certain basophil threshold. If not, users might encounter confusion when this occurs
- Re point 32 do you mean "six cell type model"?

Reviewer #2 (Remarks to the Author: Overall significance):

The authors have adequately addressed my comments.

Author rebuttal, second version:

We thank the reviewers for their invaluable input. We have addressed the minor points, and we hope those are now suitable for publication.

REVIEWERS' COMMENTS:

Reviewer #1 (Remarks to the Author: Overall significance):

Amendments are satisfactory and the manuscript has been significantly improved. A few minor comments:

- Re "Gating processes" can you explain/provide a reference?

A/ We have now included a reference (Staats et al. 2019) of the gating protocols that explains why this could occur.

-Explain TEMRA & other acronyms (important to explain every acronym)

A/ Thank you for pointing this. We have added TEMRA (T effector memory cell expressing CD45RA isoform) and reviewed that all acronyms were accounted for.

-Provide references in TEMRA/NKT section.

A/ We have added a reference (Abdelsamed et al. 2017) related to the epigenetic differentiation in the CD8T memory compartment.

-Re point 27 consider to at least include an explanation on GitHub, or flag when the signal is above a certain basophil threshold. If not, users might encounter confusion when this occurs

A/ We have now included a note for the users in the manual of the library, explaining that “unexpected variations in the cell composition, e.g., high levels of basophils >5% could point towards technical issues or unexpected non-homeostatic conditions. Always check the estimations to corroborate that the values are within a normal range for your samples.”

-Re point 32 do you mean “six cell type model”?

A/ Thank you for pointing this. The sentence was amended to “We compared the performance of cell estimates obtained applying our previously developed six model cell-type (available in Bioconductor as “FlowSorted. Blood.EPIC”).”